

**Dynamics of Snow and Glacier Cover in the Upper Karnali Basin, Nepal: An Analysis of**

**Its Relationship with Climatic and Topographic Parameters**

**Motilal Ghimire[1]***, Dibas Shrestha[2], Raju Chauhan[3], Amrit Thapa[4], Til Prasad Pangali Sharma[5],

Krishna Prasad Sharma[6], Sher Bahadur Gurung[6], Sundar Devkota[7], Prabin Bhandari[8], Sikesh

Koirala[7], Yanhong Wu[9], Niroj Timalsina[6], and Jeevan Kutu[6]

[1] *Corresponding Author:* Tribhuvan University, Central Department of Geography, Kathmandu,

Nepal. Email: motighimire@gmail.com

[2] Tribhuvan University, Central Department of Hydrology and Meteorology, Kathmandu, Nepal

[3] Tribhuvan University, Central Department of Environmental Science, Kathmandu, Nepal

[4] University of Alaska Fairbanks, Fairbanks, **AK,** USA

[5] Tribhuvan University, Nepal Mountain Academy, Kathmandu, Nepal

[6] Tribhuvan University, Central Department of Geography, Kathmandu, Nepal

[7] Department of Survey, Government of Nepal, Kathmandu, Nepal

[8] George Mason University, Fairfax, **VA**, USA

[9] Institute of Mountain Hazards and Environment, Chinese Academy of Sciences, Chengdu,

China

**Abstract**

Snow and glacier cover in the Upper Karnali Basin (UKB) are crucial freshwater reservoirs that

support downstream ecosystems and human populations. This study uses remote sensing and

GIS data **from various sources, including MODIS**-derived land surface temperature and ERA5

reanalysis **climate datasets,** to analyze snow cover dynamics from 2002 to 2024. The results

show a significant decrease in snow-covered area (SCA), with an annual decline of
**approximately** 3.99 km². Seasonal variations indicate the most significant reductions during the
monsoon period (July–September), when rising temperatures accelerate snowmelt. The analysis
also **identifies a** strong negative correlation between snow cover and temperature (r = -0.59 to -
0.77, p < 0.05), with warming trends disproportionately affecting **mid- to high-elevation** zones
(3000–5000 m a.s.l.). Glacier basins exhibit consistent retreat, with the mean glacier area
**decreasing** from 119.05 hectares in 2000 to 100.47 hectares in 2023, highlighting the impact of
climate change. Additionally, snowline analysis shows upward migration, with the 10th-
percentile snowline rising at approximately 5.16 m/year, **indicating** progressive snow loss at
lower elevations. Given the current warming trends (~0.0643°C/year above 5000 m a.s.l.), the
UKB could experience a decline in glacier area by 47–69% and snow-covered area by 19–30%.
These findings **highlight** the vulnerability of the UKB's cryosphere to climate change,
**emphasizing the need for** adaptive water resource management **strategies to** mitigate impacts
on hydrology, agriculture, and regional water security.
**Keywords:**  Snow and glacier, Karnali, Himalayas, Remote sensing, Climate change, Elevation-
dependent **warming**, Snowline

# 1. Introduction

Snow and glaciers in the mountains serve as freshwater **reservoirs**. Their meltwater provides a consistent supply to rivers and downstream ecosystems (Immerzeel et al., 2020; Wester et al., 2019; Pritchard, 2019). The meltwater from Himalayan ice and snow supports the livelihoods of millions across Nepal, India, and China by supplying drinking water, irrigation, hydropower, and ecosystem services (Bolch, 2007; Bookhagen and Burbank, 2010). Therefore, a decline in snow and glacier extent threatens water availability, food security, and sustainable development in these regions (Krishnan et al., 2019).

Furthermore, snow and glacial ice regulate regional and global climates by reflecting solar radiation, thereby contributing to the Earth's energy balance and influencing local weather patterns (Xu et al., 2009). Seasonal meltwater sustains ecosystems that provide habitats for numerous animal and plant species in mountainous regions. Consequently, changes in snow cover and glaciers can disrupt these ecosystems entirely (Wester et al., 2019). On both local and regional scales, variations in the amount of snow and ice can contribute to changes in sea level, affecting coastal areas (Forster et al., 2021; Mimura, 2013).

Snow-covered peaks and glaciers are major hubs for adventure, religious, and nature-based tourism (Anup, 2017; Nyaupane and Chhetri, 2009). Being sensitive to climate change, changes in their size and volume not only serve as visible indicators of broader climate trends but also directly threaten the tourism economy they support (Elsasser and Bürki, 2002).

A comprehensive understanding of cryospheric transformations is essential for accurate hydrological forecasting, assessing cryospheric hazards, and developing effective adaptation strategies. Historically, monitoring snow and glacier dynamics in the remote Himalayan regions

was limited by a scarcity of **in situ** observations. Since the 1970s, advances in satellite remote
sensing have revolutionized large-scale cryospheric assessments (Kääb et al., 2012; Muhammad
and Thapa, 2020). The synergistic integration of satellite-derived data with sophisticated climate
models and targeted ground-based measurements has subsequently **enabled an** improved
understanding of snow and glacier mass balance changes, their resultant hydrological impacts,
and spatiotemporal variability (Bajracharya et al., 2014; Bolch et al., 2012; Gurung et al., 2017;
Kääb et al., 2012; Krishnan et al., 2019; Kulkarni et al., 2021). Collectively, these studies
demonstrate substantial snow and glacier loss across the Himalayas, altering river discharge
seasonality and water resource availability.
Extensive research on glaciers, glacier lakes, and glacier lake outburst floods (GLOFs) in Nepal
has been conducted (Bajracharya et al., 200**9**; Hall et al., 2002; Kääb et al., 2005; Shrestha et al.,
2012). However, these **studies have** disproportionately focused on the central and eastern
Himalayas. The mid-western and far-western regions remain underrepresented due to their
remoteness and limited accessibility (**Ghimire et al., 2025a**; Khadka et al., 2024). Although
global and regional glacier **inventories that** specifically **address** high-resolution ($\leq 30$ m)
glacier cover are limited (Bajracharya et al., 2014; Bolch et al.), analyses of elevation-dependent
warming (EDW) and trend assessments are also scarce in the Himalayas (Pepin et al., 2015;
Pepin et al., 2022; Desinayak et al., 2022). Furthermore, integrated studies linking glaciers,
glacier basins, and snow cover to climate remain underexplored.
Bridging this gap is crucial for understanding cryosphere dynamics and their impacts on
hydrology, hazards, and livelihoods in western Nepal. The Karnali Basin, Nepal's largest river
basin (approximately 40,780 km² upstream of the Chisapani gauge station) and home to about
2.5 million people (CBS, 2021), exemplifies this need. Its rivers, fed by snowmelt, provide
essential dry-season water for irrigation, drinking, and hydropower. Despite its ecological
significance, the basin's cryospheric behavior remains poorly documented.
Findings from studies conducted in the central and eastern Himalayas, the Indian Himalayas, and
the Tibetan Plateau cannot be universally applied to the Karnali Basin due to differences in
climatic regimes and geographical settings. Understanding the impacts of cryosphere changes on
water resources requires research specific to the Karnali Basin. Integrating MODIS data, which
offers high temporal resolution, with Landsat data, known for its high spatial resolution, will
**improve** our understanding of snow and glacier changes and their relationships with topography,
glacier basins, and climate.
Against this backdrop, the specific objectives of this study are as follows:
1. Quantify the temporal variations in snow and glacier cover in the Upper Karnali Basin from
2000 to 2024 using multi-sensor remote sensing data.
2. Determine the influence of climatic drivers, such as rising temperatures and shifts in
precipitation, on the cryospheric dynamics, including the upward migration of the snowline.
**2. Study Area**
The Upper Karnali Basin (UKB) is a transboundary **catchment area** extending from 28.64° to
30.68° N latitude and 80.64° to 83.54° E longitude, covering 22,577 km². This region accounts
for more than 50% of the entire Karnali Basin **upstream of the** Chisapani gauge station (225 m
above sea level). It includes about 66% of the basin's glacierized area (Bajracharya et al., 2011;
Ghimire et al., 2025a). The UKB i**ncludes** the Humla Karnali (partly within Tibet, China), Mugu
Karnali, Kawari, and Tila Nadi sub-basins (see Fig. 1).
The elevation ranges from 340 meters to 7,030 meters, with an alpine zone above 4,000 meters
extending across the Middle Mountains, High Mountains, High Himalaya, and the Tibetan
Plateau. These regions encompass the geological units known as the Lesser Himalaya, Higher
Himalaya, and Tethys Himalaya (LRMP 1986; Dhital 2015). This topographic and **lithological**
diversity significantly influences climatic gradients and cryospheric processes.
The climate varies from polar tundra in the glacier regions to subtropical, temperate, and cold
climates below 4,000 meters. Mean annual temperatures range from 27 °C to -12 °C, while
precipitation varies from 250 mm in rain-shadow areas to approximately 1,900 mm annually on
the **rain-bearing** slopes. The cryosphere extends across both rainy and rain-shadow regions,
influencing the distribution and mass balance of snow and glaciers.
The Upper Karnali Basin features a diverse landscape of snow-covered glaciers, valleys,
permafrost, alpine meadows, and forests, supporting a rich variety of flora and fauna. It
represents a cultural blend of Khas and Tibetan traditions and is an emerging tourist destination,
including a stop on the Kailash Mansarovar pilgrimage route. The basin has an estimated
population of approximately 816,941 people, with a density of 36.2 persons per square
kilometer, residing in 4,395 settlements, primarily below 4,000 meters in elevation. The Human
Development Index in the area is 0.49, which is below the national average.
Due to its climatic, geological, and cryospheric diversity, the Upper Karnali Basin represents the
broader Himalayan environment. It serves as an ideal natural laboratory for studying spatial
variations in snow- and glacier-covered areas, elevation-dependent warming, and hydro-
cryospheric changes across far- and mid-western Nepal.
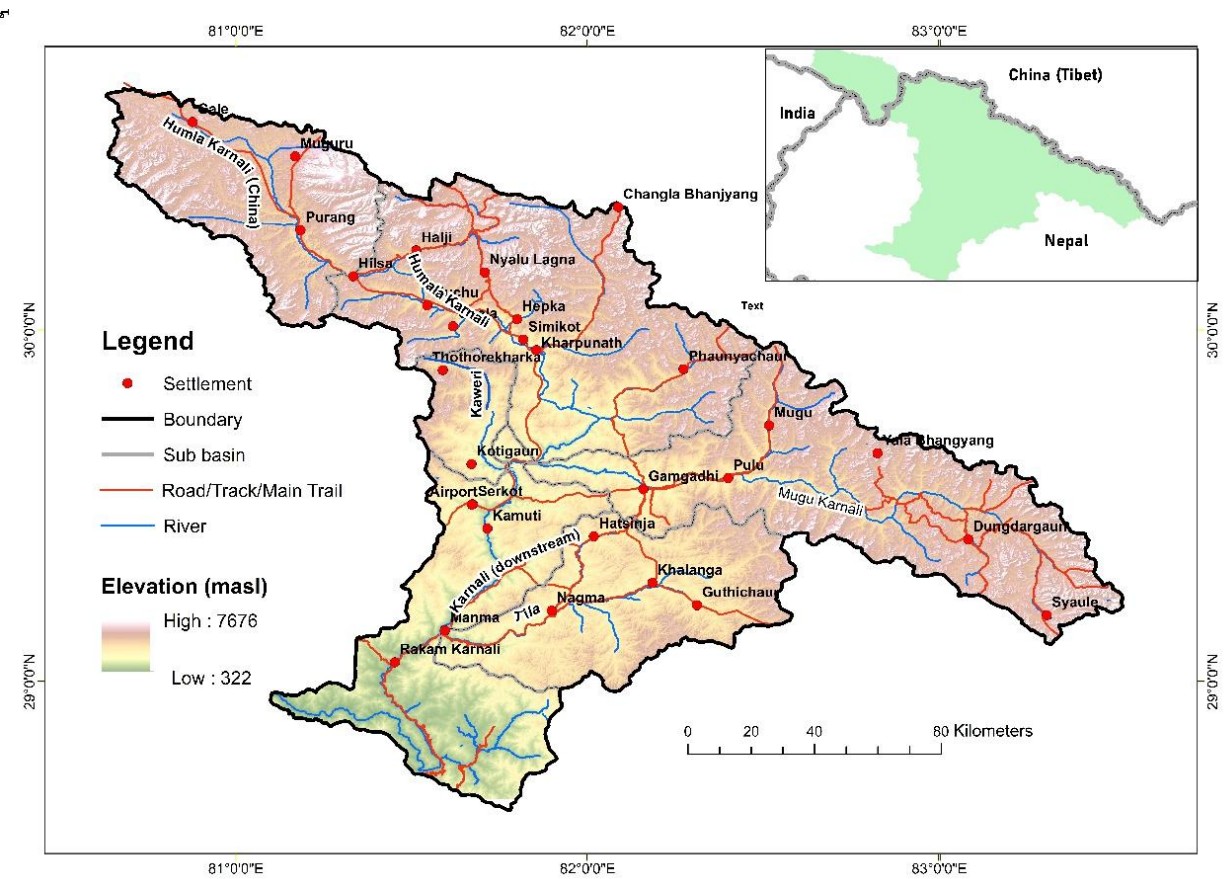
**Figure 1.** Location of the Upper Karnali Basin.

## 3. Data Sources, Methods, and Limitations

This study treats snow and glacier cover as a unified cryospheric component because of their
analogous functional roles. It analyzes cryospheric dynamics using remote sensing techniques.
Satellite imagery was processed to generate time-series data on snow and ice cover, derive land
surface temperatures, and map glacier basins.

### 3.1. Snow Cover Mapping

We mapped snow cover in the Upper Karnali Basin using Google Earth Engine (GEE) and
imagery from Landsat 5 TM, Landsat 7 ETM+, and Landsat 8 OLI. For the period preceding the
Scan Line Corrector (SLC) failure, we used only Landsat 7 ETM+ images (2002–2003). For
subsequent years, we utilized data from Landsat 5 TM and Landsat 8 OLI. To ensure high data
quality, we selected only scenes with less than 30% cloud cover see **(Sect. S1 and Fig. S1 in the**
**Supplement)**.
We preprocessed all Landsat images by masking clouds using the Quality Assessment (QA)
bands–pixel_qa for Landsat 5 and 7, and QA_PIXEL for Landsat 8. Next, we calculated the
Normalized Difference Snow Index (NDSI) using the green and short-wave infrared (SWIR)
bands (Hall et al., 2002; Gorelick et al., 2017) and applied a threshold of NDSI > 0.4 to identify
snow pixels. To reduce confusion between snow and vegetation in mixed or forested terrain, we
also calculated the Normalized Difference Vegetation Index (NDVI) and excluded pixels with
NDVI > 0.2 from the snow classification, following the approach of Rittger et al. (2013). Finally,
we exported the resulting snow cover maps as GeoTIFF files for overlay and **sub-basin** and
micro-basin analyses.
To supplement the Landsat observations, we processed MODIS 8-day composite snow-cover
products (MOD10A2) using Google Earth Engine (GEE). The MOD10A2 algorithm employs a
maximum snow-extent compositing method over each 8-day period (Parajka and Blöschl, 2008),
which minimizes cloud contamination and produces a spatially continuous dataset for analyzing
seasonal and interannual variability in snow cover. Although this **approach loses** daily temporal
resolution, the 8-day composite effectively smooths out short-lived cloud effects, providing a
more stable dataset for trend analysis.
After processing the imagery, we executed a Python script within the Google Earth Engine
(GEE) environment to automate the download and organization of snow cover data. The script
aggregated MODIS-derived snow extent by season, **sub-basin**, and elevation band (derived from
the SRTM DEM).
The year was divided into four distinct three-**month periods**: January–March (Peak
Accumulation), April–June (Major Ablation), July–September (Monsoon Ablation), and
October–December (Early Accumulation). This division was explicitly chosen to capture the
hydrological phases of snow accumulation and melting while minimizing cloud contamination
during the monsoon season (Hunt et al., 2025; Khatiwada et al., 2016; Kulkarni et al., 2010). The
resulting structured snow dataset served as the main input for analyzing snow cover trends,
elevation-dependent variability, and hydrological differences among sub-basins.
We describe the methods for spatial resolution harmonization and accuracy assessment between
Landsat and MODIS datasets (**see Sect. S2 and Tables S1–S3 in the Supplement**). Despite
these refinements, persistent monsoon cloud cover continues to limit optical remote sensing in
the Himalayas, often **leading to** underestimation of snow-covered areas and uncertainties in
seasonal trends.
Elevation bands were defined using the SRTM DEM and categorized into 200-meter intervals,
ranging from ≤2000 m to ≥6500 m. Zonal statistics were applied to extract the frequency of
snow cover for each elevation band and sub-basin. **The snow-covered area** was calculated using
a threshold-based binary mask. The results were aggregated into a structured dataset, revealing
seasonal snow distribution and variations across elevation zones and watersheds, thereby
facilitating hydrological analysis.

### 3.2. Land Surface Temperature Data and Validation

We also downloaded land surface temperature (LST) **data** at 1 km resolution from the
Application for Extracting and Exploring Analysis Ready Sample (AppEEARS) platform.
AppEEARS is a NASA-supported platform developed to facilitate easy access, subset into
specified areas, and analysis of climate and environmental **data** (Wan et al., 2015). MODIS Land
Surface Temperature (LST) data have been reliably used to determine surface temperature
patterns in areas where ground observations are scarce, **particularly** in rugged mountainous
regions. Several studies have confirmed their accuracy, **reporting** average biases of less than 1.5
K and high correlations ($R^2 > 0.9$) with on-site measurements (Duan et al., 2019; Yu et al., 2011;
Zhao et al., 2019), demonstrating their appropriateness for analyzing elevation-related warming
trends in the Himalayas. We also obtained temperature and precipitation records, including
maximum and minimum values, from the Department of Hydrology and Meteorology (DHM),
Government of Nepal, **as well as** from open-access reanalysis datasets such as ERA5.
**Temperature data** (measured at 2 m above ground)  were compared with MODIS LST; the
results of this comparison are discussed in **Sect. 4.** Due to the 1 km spatial resolution of the
MODIS product, the analysis of time series data reflects area-averaged temperature trends rather
than in situ measurements at individual stations.

### 3.3. Delineation of the Glacier Basin and Glacier Data

The boundaries of glacier basins were delineated to assess changes in glaciers and snow cover
fractions within glacier-drained areas. Glacier basins include trunk glaciers, tributary glaciers,
and surrounding slopes nourished by moving ice and snow. Their boundaries are topographically
defined, with the lower boundary terminating at the terminus of the main glacier. This
delineation process involved multiple steps to ensure accuracy.
Initially, the Glacier Inventory map referenced earlier served as a fundamental resource. High-
resolution imagery and ESRI's topographic maps in ArcGIS 10 and later versions provided
detailed spatial data. A **12.5-meter** DEM was used to extract drainage networks, produce contour
lines, and generate hillshade maps, enhancing the visualization of divides between glacier basins.
These components were essential for accurately identifying glacier termini and delineating
glacier head basins. This integrated approach, combining topographic analysis, remote sensing,
and geospatial techniques, enabled precise delineation of glacier basins for comprehensive
evaluations of snow cover fraction.
The time series glacier data compiled by **Ghimire et al. (2025)** were included in this study. The
lead author of the current manuscript **also contributed to that research** paper. In summary, we
mapped glacier polygons for the years 2000, 2010, and 2023 using high-resolution imagery from
Google Earth, Bing Maps, and RapidEye 2023 to maintain temporal consistency. Snow and
glaciers were identified based on their bright characteristics, smooth textures, and shadows cast
by adjacent terrain. Landsat composites (both true and false color) and the Normalized
Difference Snow Index (NDSI) enhanced the visibility of snow and ice, while altitude and
topographic data derived **from DEM highlighted** potential glacier regions. Outlines from the
Randolph Glacier Inventory (RGI) (Pfeffer et al., 2014) and ICIMOD (Bajracharya et al., 2011)
served as references, while ground-truth and additional data helped validate the findings. This
comprehensive approach ensured precise delineation.

### 3.4. Limitations and Validation

A key limitation of this research is that optical remote sensing is significantly affected by cloud cover, particularly during the monsoon season (Hall et al., 2002; Gafurov and Bárdossy, 2009). Frequent cloudiness often restricts the availability of clear Landsat images, leading to an underestimation of snow cover and potential inaccuracies in the spatial and seasonal assessment of snow patterns. In this study, cloud-free images were primarily available from January to March and October to December in most Upper Karnali sub-basins. Nevertheless, all four seasons were analyzed for microglacier basins where suitable data **were available.**

To address these issues, we used MODIS MOD10A2 data, which provide higher temporal resolution (8-day composites at 500 m) compared to Landsat's 16-day revisit cycle and 30 m spatial resolution. This multi-sensor strategy enhances temporal continuity and minimizes data gaps caused by clouds; however, **the** results should still be interpreted cautiously (Maskey et al., 2011a; Parajka and Blöschl, 2008).

The scarcity of high-altitude temperature stations necessitated the use of MODIS land surface temperature (LST) data at a 1 km resolution, representing daytime skin temperature at approximately 10:30 A.M. local time. This skin temperature was compared with in situ air temperature measurements taken at 2 meters above ground from four stations: Jumla (2,300 m), Simkot (2,800 m), Guthi Chaur (3,080 m), and Rara (3,048 m). Correlations varied by site and season–strongest at Jumla (up to 0.85), moderate at Guthi Chaur, and weakest at high-altitude, snow-covered sites such as Simkot and Rara ($-0.18$). MODIS LST performs well in clear, snow-free areas but requires adjustments at higher elevations. Differences arise from factors including resolution, spatial averaging, land-cover heterogeneity, and surface–air temperature contrasts.

Validation studies further confirm its reliability for analyzing high-mountain temperatures in
regions where in situ data are limited (see Duan et al., 2019; Yu et al., 2011; Zhao et al., 2019).
**4. Result**
**4.1. Snow or Ice cover Trend and Variability: Annual and Seasonal**
The total snow cover across the Upper Karnali Basin (22,546 km²) from **2002 to 2024** averages
872 km², with a standard deviation of 147 km², indicating moderate variability (Table 1 and
Figure 2). The minimum recorded snow cover is 514 km²; about 25% of the observations are at
or below 777 km². The average **snow-covered** area from **January to March** is 1,528 ± 333 km²,
followed by **April to June** (881 ± 212 km²) and **October to December** (862 ± 373 km²),
respectively. July to September shows the lowest snow cover area, i.e., 169 ± 38.3 km².
Snow cover data reveal significant year-to-year changes in every quarterly season, with varying
directions and magnitudes of trends, **as demonstrated by correlation analysis**, the Kendall tau
test, and **Sen's slope estimator**. The annual average **snow-covered area** (SCA) shows a
decreasing trend, **although it is not statistically significant** (p = 0.535). **Sen's slope estimates** a
loss of approximately 3.99 km² per year, **indicating** a gradual decline in snowpack **over the past**
two decades. Seasonally, the July–September period exhibits a gentler trend compared to
October–December; however, **due to its** much lower inter-annual variability, this period exhibits
the statistically significant steepest decline in snow cover (Sen's Slope = -2.87, p = 0.001) (Table
1). This period is characterized by **snow ablation**, as the summer monsoon brings warmer
temperatures. In mid-latitude regions, precipitation occurs more as rain than snow, resulting in
accelerated snowmelt. While January–March shows a decline (Sen's slope = -8.63 km/year), it **is**
**not statistically significant** (p = 0.523), suggesting year-to-year winter variability in snowfall or
early melt. Similarly, no significant trends were detected from **April to June**. Interannual
variability is evident, with peaks and lows in snow and ice coverage (Figure 2). Episodic snow
coverage was observed in 2015, 2020, and 2022 (**January–March**); 2015 and 2019 (**April–**
**June**); and 2009 and 2021 (October–December), indicating **anomalous** years **of** heavy episodic
snowfall events. However, these anomalies do not **offset** the long-term declines. Compared to
**seasonal variability**, annual snow **coverage shows** relatively **low interannual variability**, with
a 16% coefficient of variation **(CoV)–the ratio** of the standard deviation to the mean.

**Table 1**. Snow cover descriptors and changes by seasons

| Descriptor | Jan–Mar | Apr–Jun | Jul–Sep | Oct–Dec | Annual avg. |
|---|---|---|---|---|---|
| Mean (km²) | 1 528.00 | 881.00 | 217.00 | 862.00 | 872.00 |
| Median (km²) | 1 569.00 | 858.00 | 210.00 | 739.00 | 886.00 |
| Std. dev. (km²) | 333.00 | 212.00 | 38.30 | 373.00 | 147.00 |
| Minimum (km²) | 1 025.00 | 503.00 | 169.00 | 340.00 | 514.00 |
| Maximum (km²) | 2 167.00 | 1 358.00 | 298.00 | 1 737.00 | 1 055.00 |
| Skewness | 0.21 | 0.47 | 0.94 | 0.51 | −0.87 |
| 25th percentile (km²) | 1 270.00 | 751.00 | 191.00 | 538.00 | 777.00 |
| 50th percentile (km²) | 1 569.00 | 858.00 | 210.00 | 739.00 | 886.00 |
| 75th percentile (km²) | 1 689.00 | 1 025.00 | 229.00 | 1 126.00 | 991.00 |
| Correlation (r) | −0.09 | −0.07 | −0.61 | −0.25 | −0.27 |
| Kendall's τ | −0.09 | 0.01 | −0.54 | −0.13 | −0.10 |
| p-value | 0.523 | 0.950 | 0.000 | 0.398 | 0.535 |
| Sen's slope (km² yr⁻¹) | −8.63 | −3.14 | −2.87 | −13.21 | −3.99 |


**Note:** Sen's slope represents the median of all possible pairwise slopes, **quantifying the** trend
(here, snow cover) over time (Sen, 1968). It provides a more reliable **long-term estimate** of
snow cover loss without being skewed by short-term anomalies (Gilbert, 1987; Yue and Wang,

281    2004).


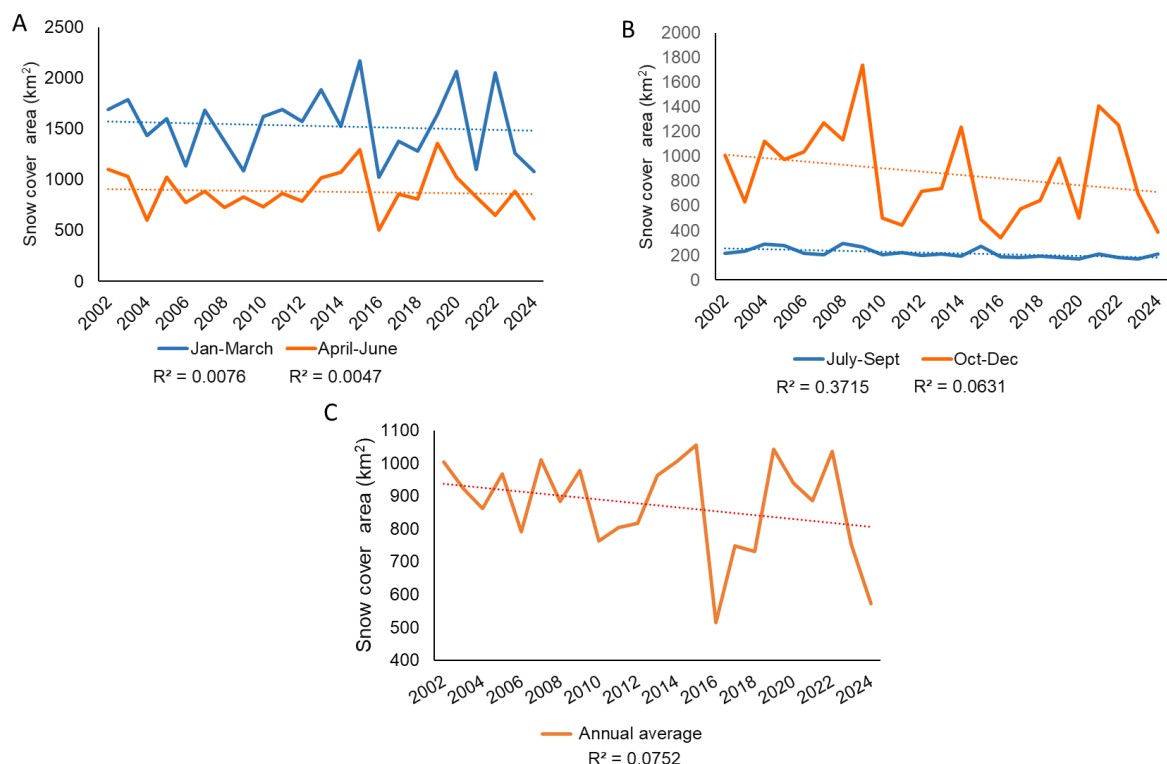


**Figure 2.** Temporal variation and trends in seasonal and annual snow-covered area (SCA) in the

Upper Karnali Basin (2002–2024). (a) Time series of SCA for January–March (orange) and

April–June (green); (b) SCA for July–September (blue) and October–December (orange); (c)

average annual SCA (orange).

**4.2. The Relation between Snow Cover, Temperature, and Precipitation**

We derived land surface temperature (LST) data for 204 locations from MODIS Terra

(MOD11A1) and Aqua (MYD11A2) **products at 1 km resolution**, processed through

AppEEARS. Precipitation data were obtained from the ERA5-Land reanalysis (~9 km

resolution) **provided by** ECMWF (Hersbach et al., 2020). These datasets, covering four distinct

seasons, were used to **analyze** temperature and precipitation trends, as well as their relationships

with snow cover trends **(Figures 3-5).**

Using correlation statistics, we found that among the 204 sampled sites, 143 locations
(approximately 70%) exhibit a positive annual temperature trend, indicating a general warming
pattern throughout the study region (Figure 3). However, statistically significant trends ($p \leq 0.1$)
were identified in only a subset of these sites, highlighting that not all observed warming trends
are statistically robust. Moreover, the warming pattern is not consistent across all seasons.
Notably, during the April–June interval, the temperature trend tends to be weaker or, in some
cases, negative. Several subsites across different seasons also demonstrate negative trends,
although the majority of locations show a positive trend (Figure 3). Elevation-related variability
in these trends is further analyzed in Figures 7–9 and Table 3.
Seasonal rainfall trends from 2000 to 2024 indicate weak to moderate increases across all
seasons, except for winter (January–March), which shows a slight downward trend ($R^2 = 0.014$)
(Figure 4). **Pre-monsoon** (April–June) rainfall **exhibits** a slight upward trend ($R^2 = 0.0119$). All
these seasons display high variability, suggesting a limited impact on snow accumulation.
Monsoon rainfall (July–September) demonstrates a more noticeable increase ($R^2 = 0.0975$),
primarily contributing to rainfall rather than snowfall. Post-monsoon (October–December)
precipitation remains low and stable. **Combined** with rising temperatures, these trends indicate a
shift toward rainfall-dominated precipitation, reduced snowfall, and earlier snowmelt,
contributing to declining snow cover and altered hydrological regimes.
The snow-covered area exhibits a strong to moderate negative **correlation with temperature**
**across all seasons** ($r = -0.59$ to $-0.77$, $p < 0.05$) (Figure 5). **Conversely, precipitation shows a**
**positive correlation with snow cover during January–March and October–December** ($r =$
$0.55$ to $0.59$, $p < 0.05$), while in the remaining seasons, it demonstrates a moderate negative
correlation. **Additionally, precipitation** and temperature are negatively correlated in winter
(**October–March**) and positively correlated in summer (**April–September**).

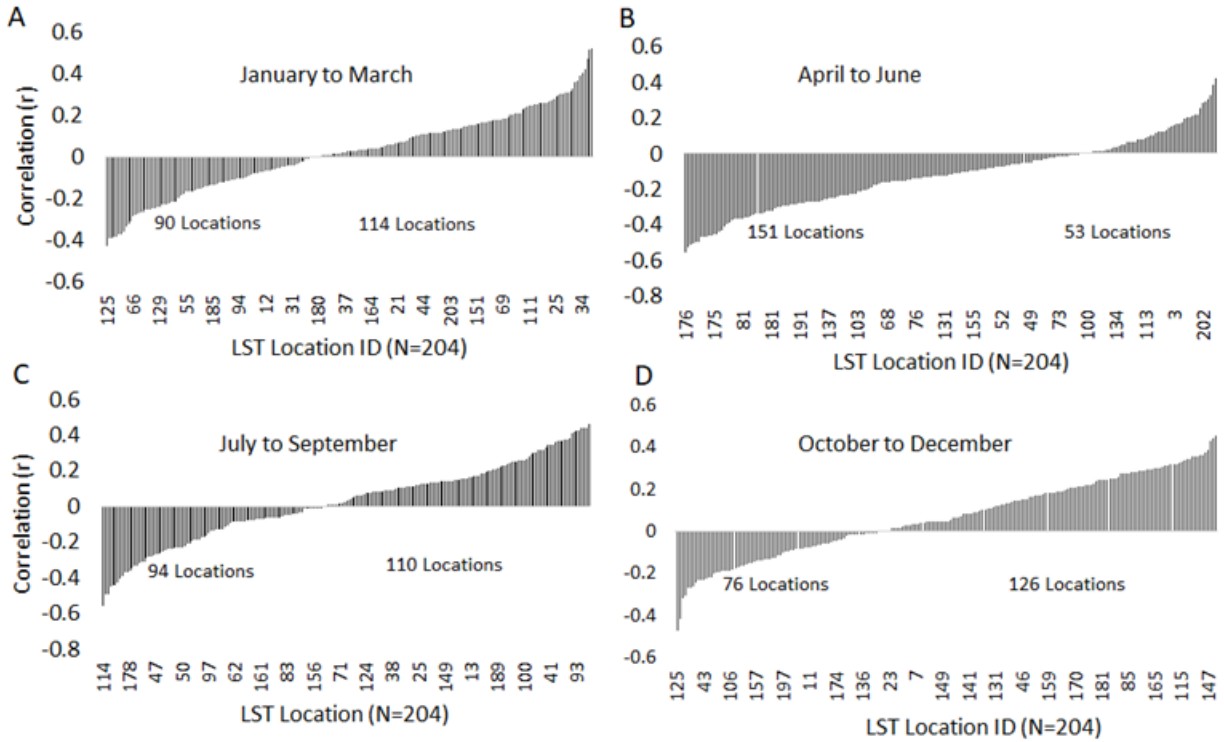


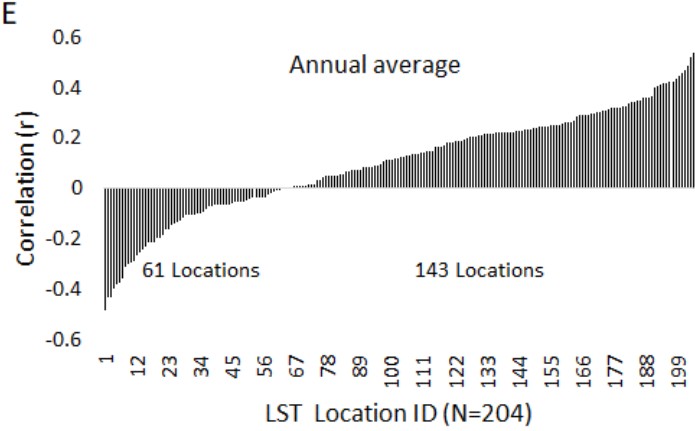


**Figure 3.** The correlation illustrates the **seasonal** (A–E) temperature trend directions at various
sites between 2000 and 2024 (Source: MODIS Terra and Aqua MOD11A2, MYD11A1,
AppEEARS). Significant correlations at **the** 90% confidence level are observed at r = ±0.364.

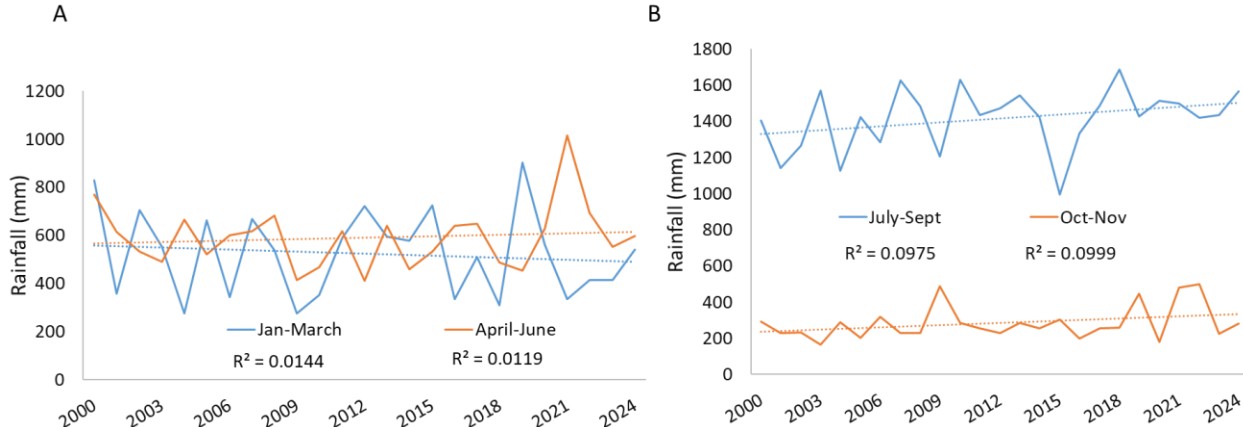


**Figure 4.** Yearly rainfall trends across various periods. Precipitation data were collected from the ERA5-Land reanalysis dataset by ECMWF (Hersbach et al., 2020), covering 204 locations over four distinct time intervals.



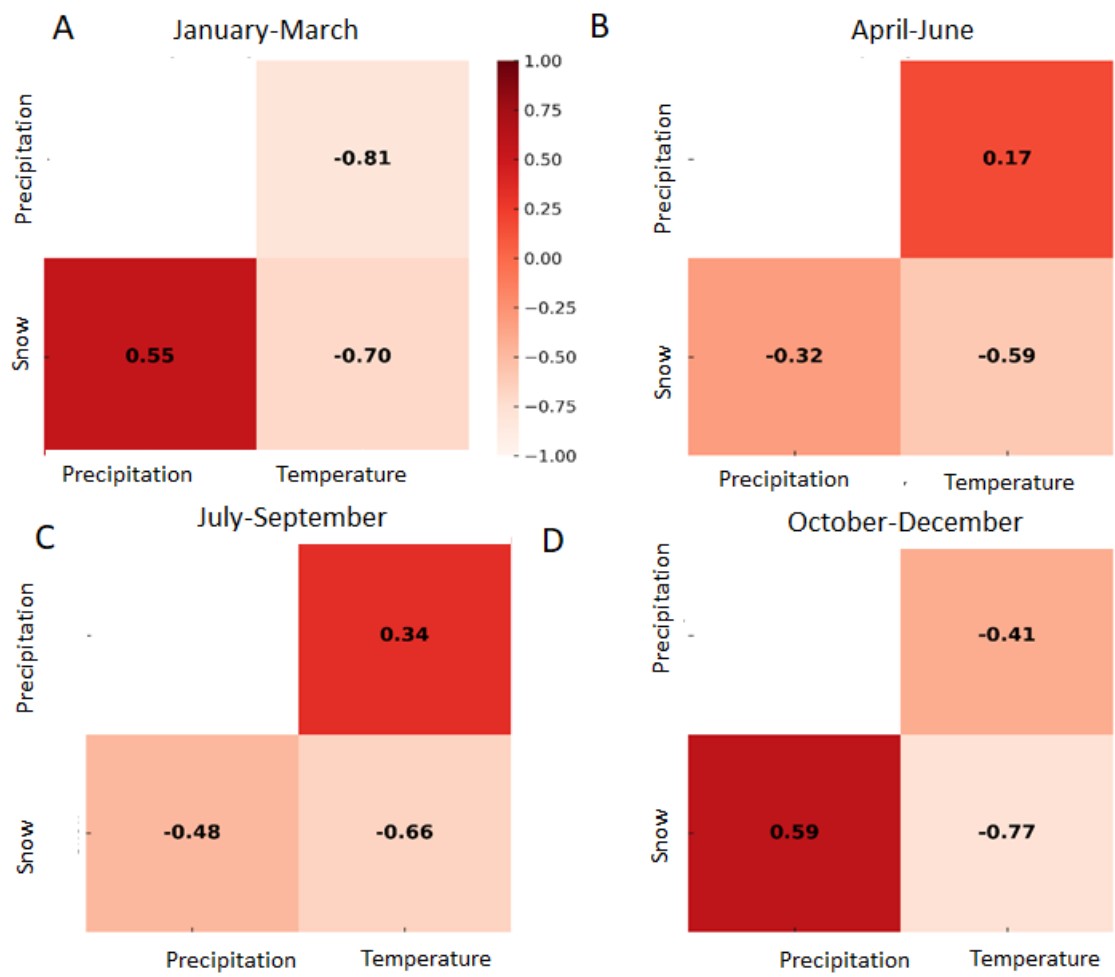

**Figure 5**. Seasonal correlation patterns among snow cover, temperature, and precipitation **over** a 22-year period, **presented** separately for each season (A–D).

### 4.3. Snow Cover Changes in Sub-Basins Using Landsat Series Data

Landsat-derived reliable snow and ice data were unavailable for the pre-monsoon and monsoon seasons due to significant cloud **cover** (as mentioned in Section 3.4). Therefore, only two **seasons: January to March** and **October to December**, were considered. These periods are characterized by snowfall **as the primary form of precipitation**, contributing to snow accumulation.

341

Examining snow cover patterns in the sub-basins of the Upper Karnali Basin (UKB) across two

seasons (January–March and October–December) reveals notable seasonal and spatial

differences (Table 2). During January–March, Humla Karnali **exhibits** the largest average snow

cover (3,336 km²), followed by Mugu Karnali (1,864 km²) and Humla Karnali (China) (1,478

km²), **while downstream areas** such as Tila and Kawari have **minimal** coverage (less than 350

km²). Significant variability in snow cover **trends** is observed, particularly in Tila and

Downstream Karnali, with a coefficient of variation (CoV) **exceeding** 50%. This high CoV

indicates inconsistent snow cover from year to year during January–March. Furthermore, this

variability is associated with a significant negative **correlation**, i.e., $r \leq -0.37$ ($p < 0.1$). Figure 6

**graphically illustrates** the temporal **trends, showing** the correlation coefficient (r) and

fluctuations in Landsat-derived snow cover for the two seasons mentioned above. The

moderately negative skewness of the temporal distribution does not affect the correlation, which

is negative for all basins, indicating a declining trend.

Conversely, the October–December season has a lower average snow cover (823 km²) and

**exhibits significant fluctuations**, with a range of 227–1,570 km² and a coefficient of variation

(CoV) of 55%. Strong variability is observed **across all basins**, particularly in Humla Karnali

(China), Tila, and Downstream Karnali. The skewness **is moderate for most basins**, except for

the Downstream Karnali. Correlation values are reliable and indicate a declining trend. **Notably,**

**despite** high variability, **Downstream Karnali** shows a statistically significant negative

**correlation coefficient** of -0.47 ($p < 0.05$) (Figure 6).


**Table 2.** Descriptive statistics of snow cover across sub-basins for two seasons (January–March and October–December), along with the time series correlation from 2002 to 2024.

| Descriptor | January - March | | | | | | | October-December | | | | | | |
| --- | --- | --- | --- | --- | --- | --- | --- | --- | --- | --- | --- | --- | --- | --- |
| | Humla Karnali (China) | Humla Karnali | Mugu Karnali | Tila | Kawari | Downstream | Seasonal average | Humla Karnali (China) | Humla Karnali | Mugu Karnali | Tila | Kawari | Karnali (Downstream) | Seasonal average |
| Mean | 1478 | 3336 | 1864 | 351 | 294 | 41.9 | 1227 | 854 | 2057 | 1442 | 332 | 204 | 48.1 | 823 |
| Median | 1420 | 3667 | 1827 | 346 | 308 | 39 | 1239 | 754 | 2159 | 1301 | 288 | 190 | 40.2 | 781 |
| Standard deviation | 501 | 597 | 645 | 184 | 86.2 | 24 | 311 | 622 | 1163 | 862 | 227 | 112 | 44.7 | 457 |
| Coefficient of variation (CoV in %) | 33.90 | 17.90 | 34.60 | 52.42 | 29.32 | 57.28 | 25.35 | 72.83 | 56.54 | 59.78 | 68.37 | 54.90 | 92.93 | 55.53 |
| Minimum | 552 | 1904 | 887 | 50.1 | 121 | 5.74 | 612 | 67.2 | 166 | 226 | 44.3 | 35.2 | 3.28 | 227 |
| Maximum | 2317 | 4009 | 3056 | 691 | 420 | 93.5 | 1642 | 2092 | 4074 | 2868 | 716 | 434 | 185 | 1570 |
| Skewness | -0.707 | -0.488 | -1.29 | -0.69 | -0.469 | -0.763 | -1.1 | 0.533 | -0.016 | 0.231 | 0.347 | 0.204 | 1.43 | 0.202 |
| Temporal correlation (r<-0.44 and r>0.44, p<0.05) | -0.16 | -0.18 | -0.10 | 0.12 | -0.37 | -0.41 | -0.14 | -0.16 | -0.07 | -0.23 | -0.17 | -0.13 | -0.47 | -0.17 |




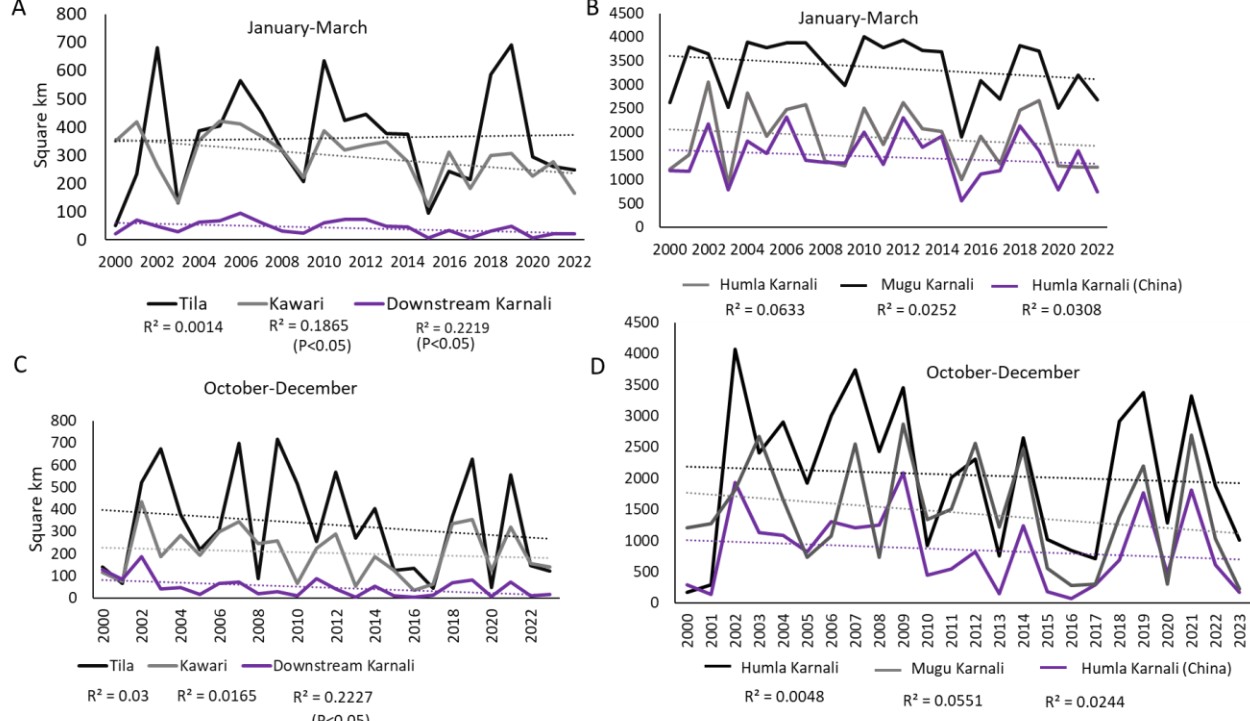


**Figure 6.** The snow cover trend in the Upper Karnali Basin varies across different sub-basins

from January–March and from October–December (A-D).

## 4.4. Snow Cover Dynamics across Elevation Zones

The dynamics of snow cover across elevation zones, categorized in 200-meter intervals from

≤2000 m to ≥6500 m, reveal remarkable elevation-dependent patterns in correlation and

**variability over time (2002–2024)** (Figure 7). Snow cover in the lowest elevation zones

**exhibits** a weak positive correlation (0.12–0.43), indicating a slight increase. However,

pronounced interannual variability (CoV ~ 41-43%) is likely driven by fluctuating temperature
and precipitation regimes (Pendergrass, 2020).
Above 2300 m a.s.l., correlations shift to weak negative values (up to 5000 m a.s.l., r = -0.05 to -
0.17), reaching peak negativity at 6100–6200 m a.s.l. (r = -0.56), indicating a significant decline
in snow cover (Figure 7). This trend aligns with the impacts of global warming, where rising
temperatures disproportionately affect higher elevations, accelerating snowmelt and reducing
accumulation (Naegeli et al., 2019; Ren et al., 2023; Shen et al., 2021). The mean snow cover
increases with elevation, showing a marked rise from 3300 to 6500 m a.s.l. or above, except
between 5000 and 5200 m a.s.l., which exhibits a gradual increase in snow cover.
Above this elevation, the mean snow cover area increases sharply, coinciding with glaciers and
permanent snow zones. In contrast, **the coefficient of variation (CoV)** rises with elevation up to
3100 m a.s.l., then declines sharply from 3100 m a.s.l. to 6500 m a.s.l. and beyond. This pattern
indicates a decrease in interannual variability **accompanied by stronger** negative correlations.
The low interannual variability reinforces the reliability of the observed declining trend in snow

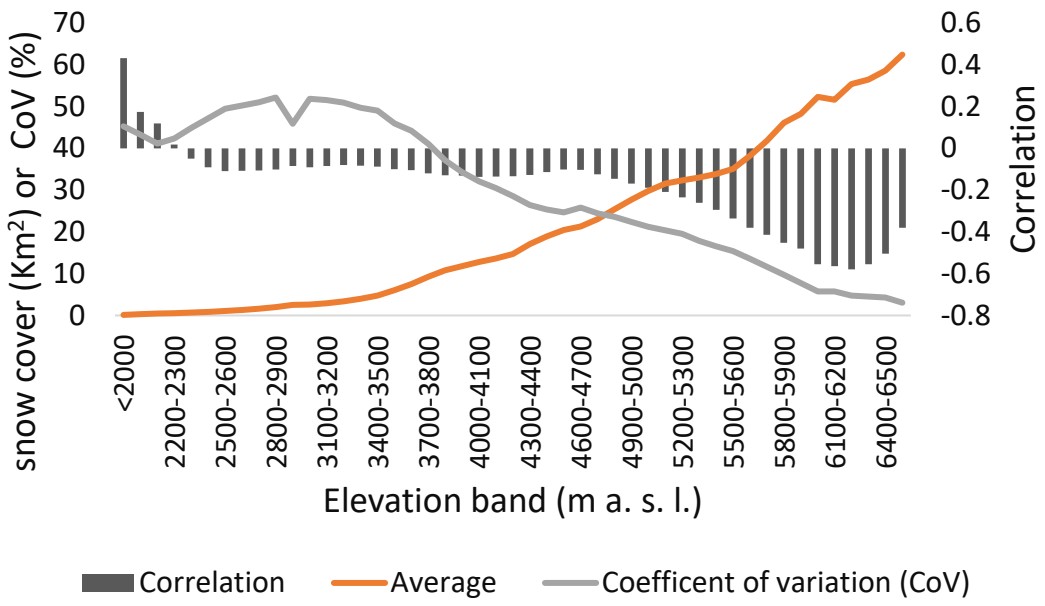

**Figure 7.** The average, coefficient of variation, and correlation of snow cover area (Source: MODIS) across various elevation bands with time (2002–2024).

To examine the relationship between temperature and snow cover, the elevation bands were regrouped into **seven broader categories: <1000 m, 1000–2000 m, 2000–3000 m, 3000–4000 m, 4000–5000 m, 5000–5500 m, 5500–6000 m,** and above 6000 m a.s.l. The temperature trend from 2002 to 2024 across these elevation bands in the Upper Karnali Basin, as **indicated** by Sen's slope (Figure 8, Table 3), shows a general increase. The highest rate of change is observed at lower elevations (<1000 m: 0.0765°C/year). Mid-elevations (**1000–2000 m:** 0.0576°C/year) and high elevations (5000–5500 m: 0.0643°C/year) also exhibit significant warming. **However, the** statistical significance (P-value) weakens at higher elevations. This warming accelerates **glacier retreat, reducing snow cover and altering river flow patterns, thereby reducing the glacier-fed water supply in the Upper Karnali Basin.**



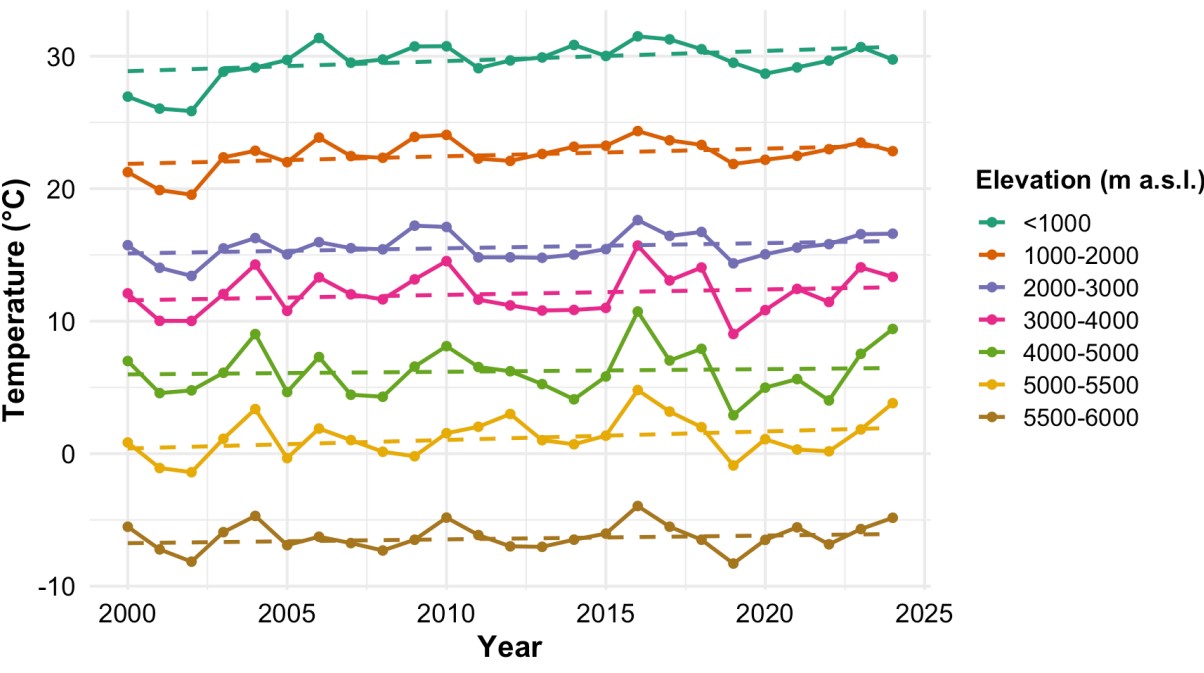


**Figure 8.** Temperature (source: MODIS) trend between 2002 and 2024 for different elevation

bands



**Table 3.** Rate of temperature change in different elevation between 2000–**2024.**

| Elevation bands (m a.s.l.) | Sen's slope (Sen, 1968) | P Value |
|---|---|---|
| <1000 | 0.0765 | 0.052 |
| 1000–2000 | 0.0576 | 0.058 |
| 2000–3000 | 0.0390 | 0.168 |
| 3000–4000 | 0.0410 | 0.528 |
| 4000–5000 | 0.0198 | 0.833 |
| 5000–5500 | 0.0643 | 0.154 |
| 5500–6000 | 0.0287 | 0.414 |

Figure 9 shows a strong negative correlation between land surface temperature and snow cover

across elevation bands in the Upper Karnali Basin. Tau values range from -0.43 to -0.79. The

correlation is strongest **between 3000 and 5000 m a.s.l.** (Tau = -0.77 to -0.79) and 5000–5500 m

a.s.l. (Tau = -0.75), with all p-values <0.01, confirming statistical significance. Even at 5500–

6000 m a.s.l. (Tau = -0.43, p = 0.00353), snow cover continues to decline. The impact is most

severe at mid-to-high elevations, where warming accelerates snowmelt and glacier retreat,

highlighting the vulnerability of the Upper Karnali Basin's hydrological balance to climate

change.





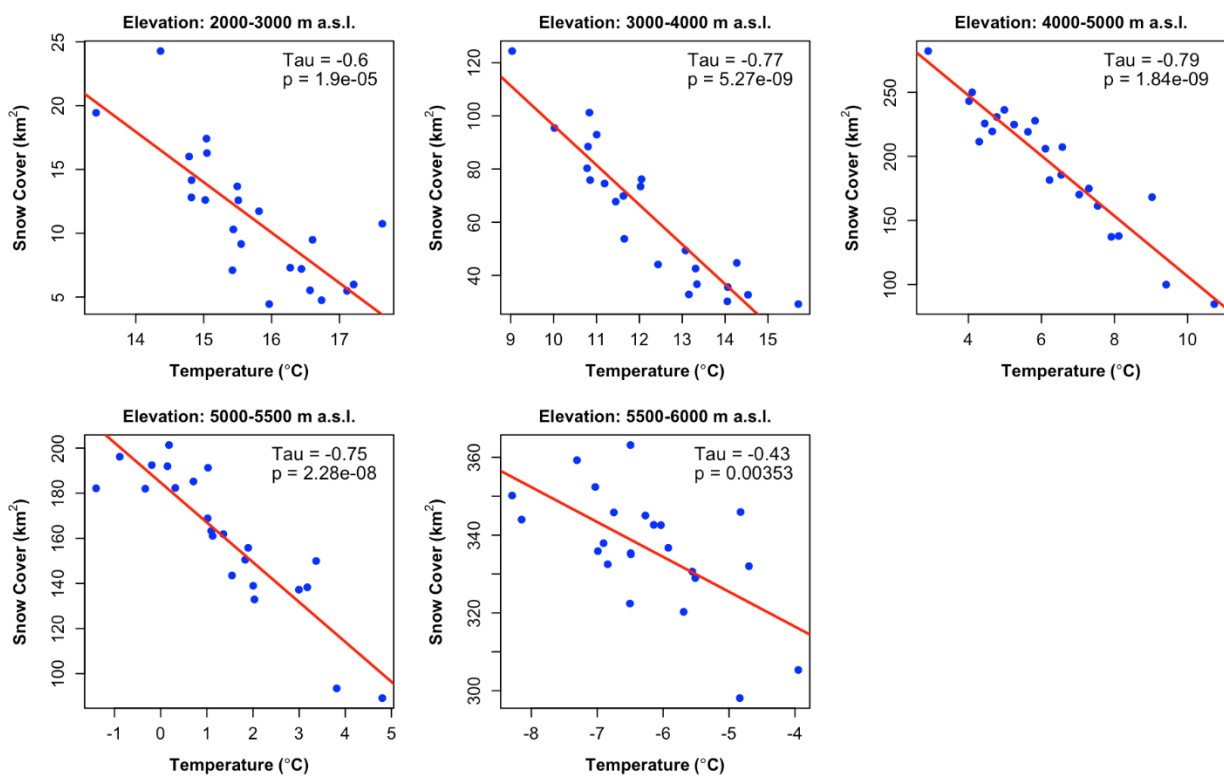


**Figure 9.** Relationship between snow cover and temperature (°C) across elevation zones in the
Upper Karnali Basin (2002–2024). The correlation (Kendall's Tau) shows a strong negative
association **at all** elevations, especially between 3000–5500 m a.s.l., where warming has
significantly reduced snow cover.

**Note:** Elevation bands below 2000 m are excluded due to minimal snow presence, high
interannual variability, and limited data reliability.

### 4.5. Snow Cover Trend in Glacier Basins (Landsat Data).

We examined **snow cover trends using Landsat data in** 735 glacier basins, each containing at
least one glacier in 2000 that was greater than 10 hectares, which are crucial for assessing glacial
status, water security, and climate change impacts (Table 4). The minimum altitude of the glacier
basin, where all tributary glaciers **converge**, was considered the outlet of the glacier basin. In
these basins, snowfall **replenishes the** ice lost to melting, contributing to glacier stability.

Reduced snow cover in the glacier basins accelerates negative mass balance, leading to glacier
retreat. These glacier basins are located at a minimum altitude above 4000 m a.s.l., with an
**average altitude** of approximately 5100 m a.s.l. **Twenty-five and seventy-five percent of the**
**basins** lie below 4800 m and 5330 m a.s.l., respectively. **In addition** to other meteorological
parameters, current temperature trends and albedo patterns play a critical role in glacier mass
balance (Dowson et al., 2020; Ye & Tian, 2022). Higher temperatures directly increase the
**snowmelt** rate, and a decrease **in the** reflectivity of solar radiation **causes** more solar energy to
**be absorbed** by the glacier surface, leading to accelerated melting. Declining permanent snow
cover in the glacier basin disrupts the glacier mass balance, affecting glacier persistence, altering
water availability, and accentuating climate-driven environmental changes.
The data reveal a significant decline in glacier area across 735 glacier basins between 2000 and
2023. The mean glacier area decreased from 119.0 **hectares** in 2000 to 100.5 hectares in 2023,
reflecting an average loss of 18.6 hectares per basin. The total glacier area shrank by 13,633.2
hectares, indicating widespread glacier retreat. The percentage of glacier area **relative to the**
**total** basin area declined from 53.23% in 2000 to 44.93% in 2023. Statistical tests show high
skewness (>3.9), suggesting that a few large glaciers dominate the dataset. The Shapiro-Wilk test
$(p < .001)$ confirms a non-normal distribution.
**Table 4.** Change in glacier area between 2000 and 2023.

| Glacier basin count (N=735) | Glacier basin Area (hectares) | Glacier area (hectares) | | Difference in glacier area (hectares) |
|---|---|---|---|---|
| | | 2000 | 2023 | |
| Median | 101.4 | 52.8 | 39.7 | -10.0 |
| Mean | 223.6 | 119.0 | 100.5 | -18.6 |

| Std. Dev | 368.1 | 187.1 | 169.9 | 27.2 |
| --- | --- | --- | --- | --- |
| Skewness | 4.6 | 4.0 | 4.0 | -4.0 |
| Sum | 164140.9 | 87379.9 | 73746.8 | -13633.2 |

The glacier area has declined significantly across all basin **orientations** from 2000 to 2023, with
basins facing **north, northwest, and northeast** experiencing the largest losses, totaling 6,126.9
hectares (Figure 10). **Glaciers on northeast, east, and south-facing slopes exhibit the highest**
**relative percentage loss.** This consistent decline across all directions underscores the ongoing
impact of climate change on the region's glacier-fed water resources.

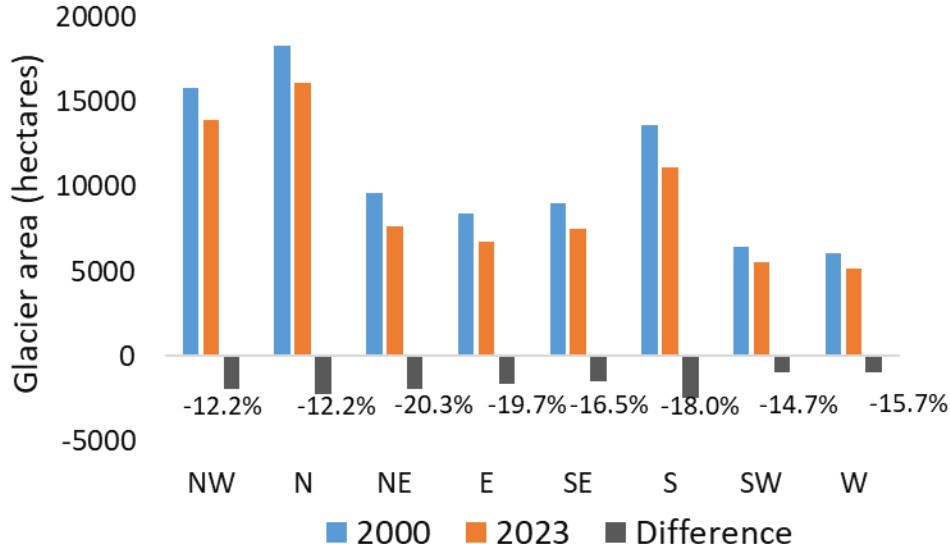


**Figure 10.** Change in glacier area in glacier basins by direction between 2000 and 2023.
Analysis of snow cover trends indicates that **approximately** 59% of glacier basins (n = 735)
exhibit statistically significant negative correlations (p < 0.05) from January to March. Among
these, basins with a **correlation coefficient** (r) less than -0.44 **account for** 16.3% of the total
(Figures 11, 12, and 13). Basins with moderate negative correlations, ranging from -0.44 to -
0.30, **represent a**bout 19% of the total. Additionally, 36% of basins show positive **correlations,**
with 3% being statistically significant and 13% displaying a moderate **correlation**. The
**prevalence** of glacier basins with negative correlations **suggests** a broader regional trend of
declining **winter snow** cover (January to March).
Similarly, **from May to July**, all 15 cloud-free glacier basins **exhibit** a declining trend in snow
cover from 2002 to 2024. Twelve of these basins **show** a moderate negative correlation ($r < -$
$0.30$). The snow **cover trend** during July to September and October to December also indicates a
decline. **Sixty-two percent** of the 70 glacier basins display a statistically significant negative
correlation ($p < 0.05$).







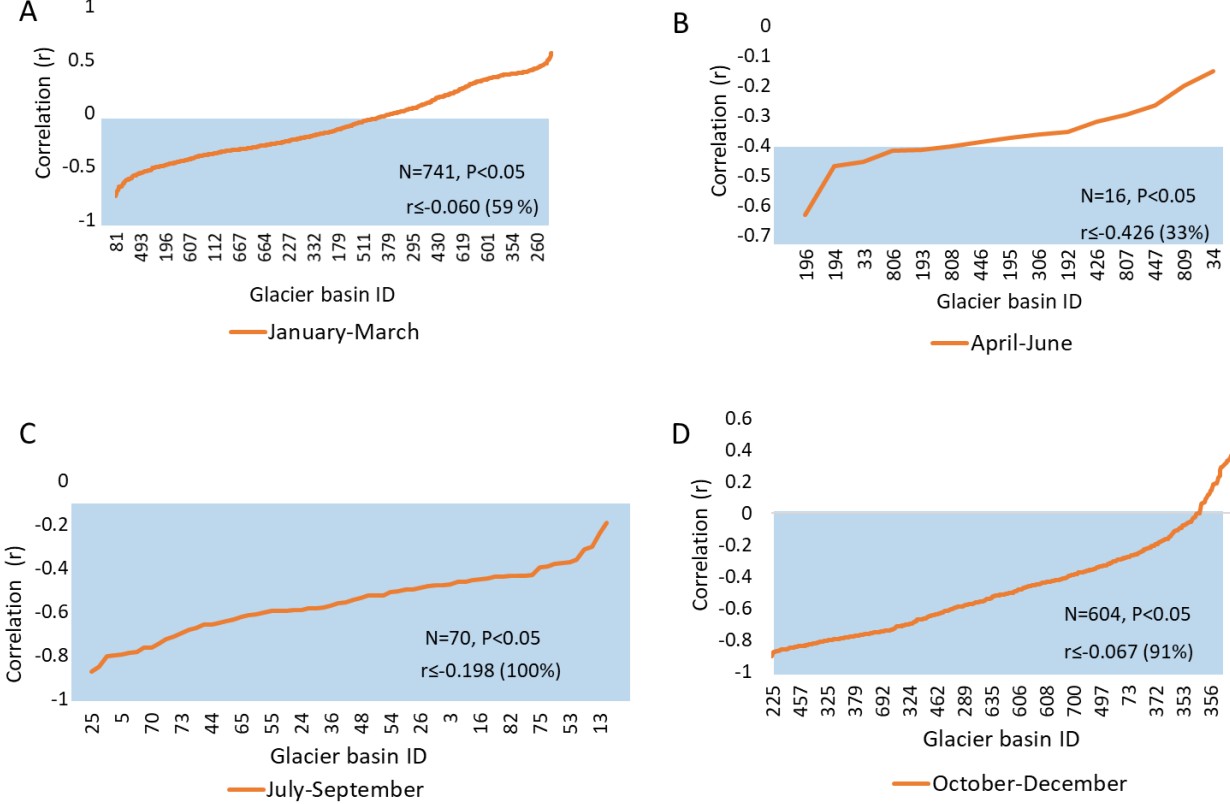

**Figure 11.** The correlation showing the snow cover change between 2002 and 2024 in different glacier basins.

The snow cover trend between July and September and **between** October and December over 22 years also demonstrated a consistent decline across all glacier basins. **Of the** 604 basins selected for analysis, **approximately** 91% showed a statistically significant negative correlation (p < 0.05), and 15% of the glacier basins exhibited a moderate negative correlation, with r values ranging from -0.47 to -0.30 (**Figures 11-13**). The snow cover in the remaining basins showed a **weak negative correlation b**ut still indicated a decline over the period.

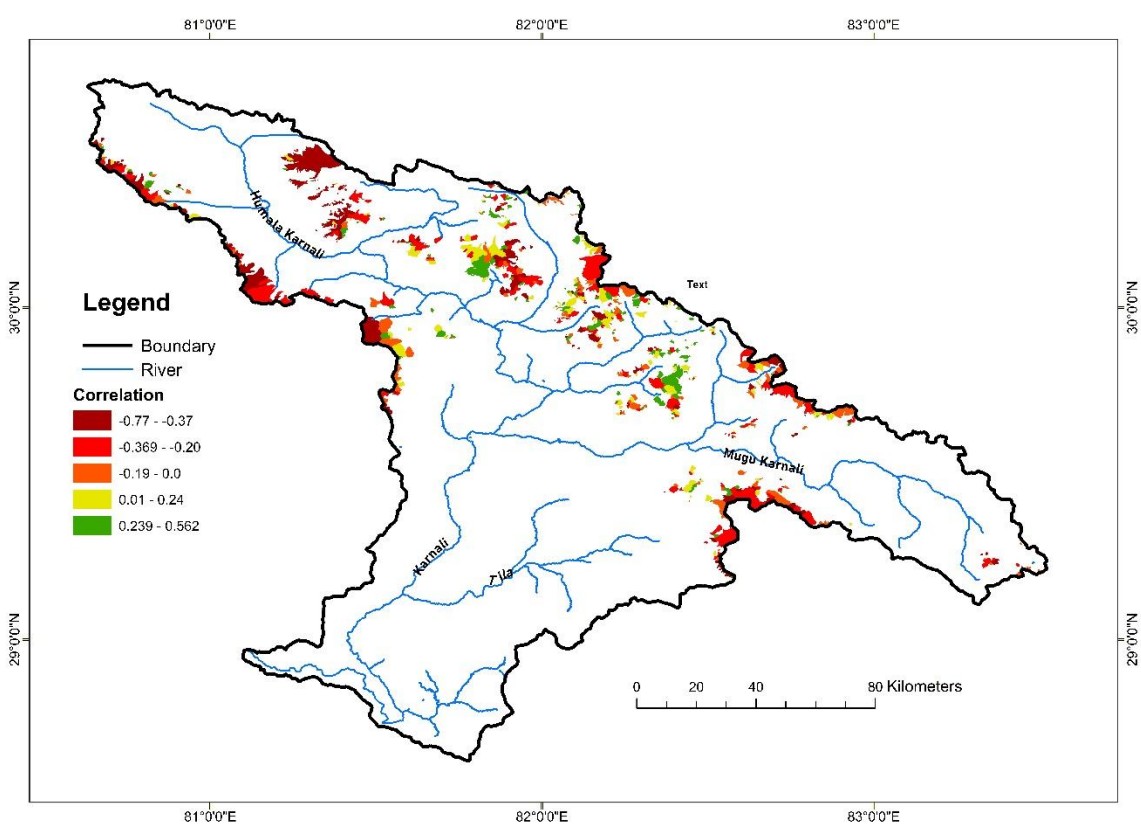

491

**Figure 12.** Snow cover trend on the Glacier Basins for January–March between 2000–2023

(Landsat 5, 7, and 8).

494

495

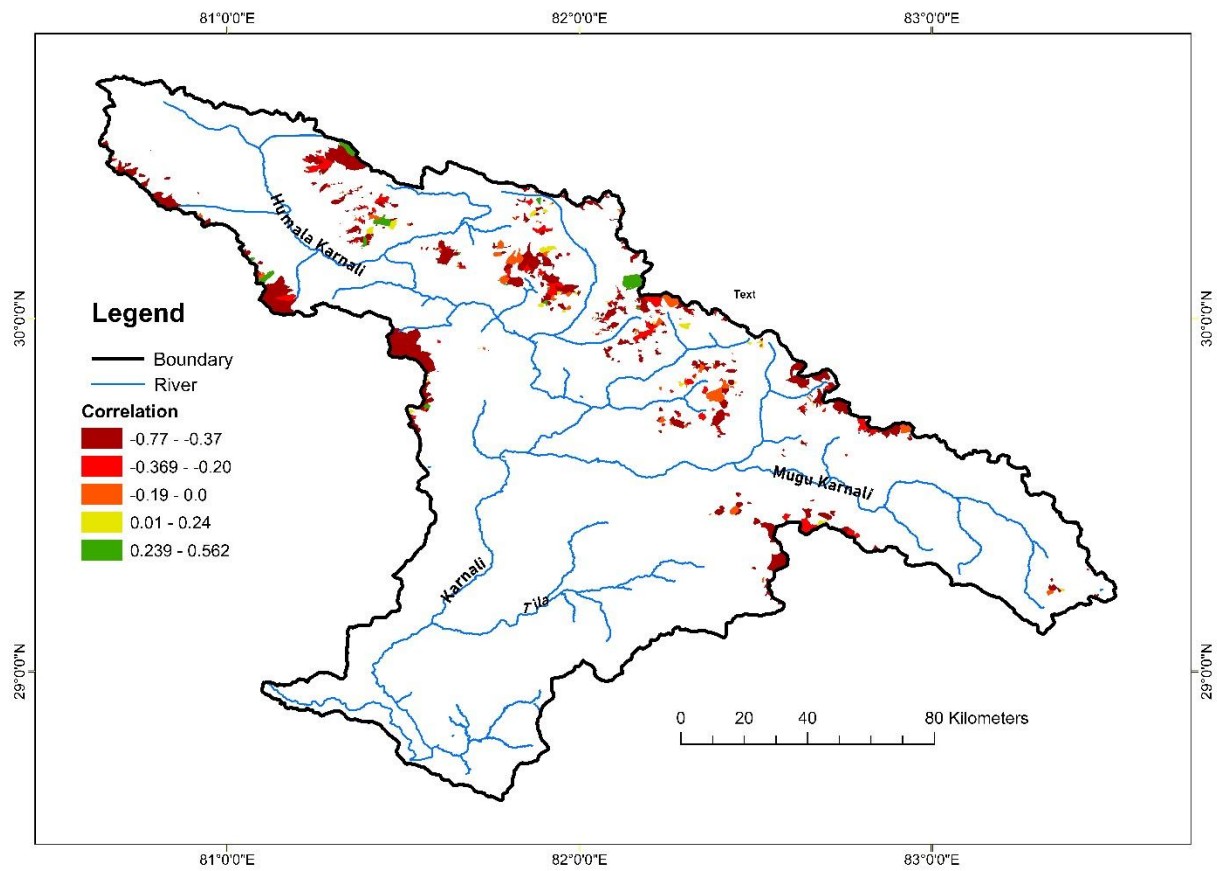

**Figure 13.** Snow cover trend on the glacier basins for October–December between 2000–2023 (Landsat 5, 7, and 8).

## 4.6. Snowline Shift across Elevations

Snow-covered areas were derived from **Landsat 7, 8, and 9** imagery by classifying snow using the Normalized Difference Snow Index (NDSI) algorithm to analyze changes in the snowline. This analysis was performed on the Google Earth Engine (GEE) platform. Snow pixels were **detected** using an NDSI threshold of > 0.4. The elevation-wise distribution of snow pixels was then calculated. To determine the minimum **elevation of the snowline** and its shift from 2002 to 2024, three statistical thresholds were applied: the 10th, 25th, and 50th percentiles of the snow cover distribution across **different** elevations.

The analysis of snowline altitude data from 2002 to 2024 reveals a significant upward trend
across all percentiles (Figure 14). The 10th percentile shows the **most significant** increase, with
a Kendall's tau of 0.2662 and a Sen's slope of approximately 5.16 m/year, indicating that the
lower snowline is rising rapidly (Table 5). The 25th percentile presents a moderate yet
statistically significant trend, with a Kendall's tau of 0.1938 and a Sen's slope of about 2.91
m/year. In contrast, the 50th percentile shows a gentler trend, with a Kendall's tau of 0.1483 and
a Sen's slope of around 1.54 m/year, **both of which remain statistically** significant ($p < 0.05$).
Collectively, these findings suggest that the snowline is shifting to higher elevations, reflecting
broader climatic changes that **impact** lower elevations more **intensely** than the median snowline
altitude.
**Table 5.** Statistical analysis of snow line altitude trends using Kendall's Tau and Sen's slope.

| Snow Line Percentile | Kendall's Tau | p-value | Sen's Slope (m/year) | Significance |
|---|---|---|---|---|
| 10th Percentile (SLA_10P) | 0.2662 | 0.00042 | 5.16 | Significant ($p < 0.001$) |
| 25th Percentile (SLA_25P) | 0.1938 | 0.01022 | 2.91 | Significant ($p < 0.05$) |
| 50th Percentile (SLA_50P) | 0.1483 | 0.04942 | 1.54 | Significant ($p < 0.05$) |


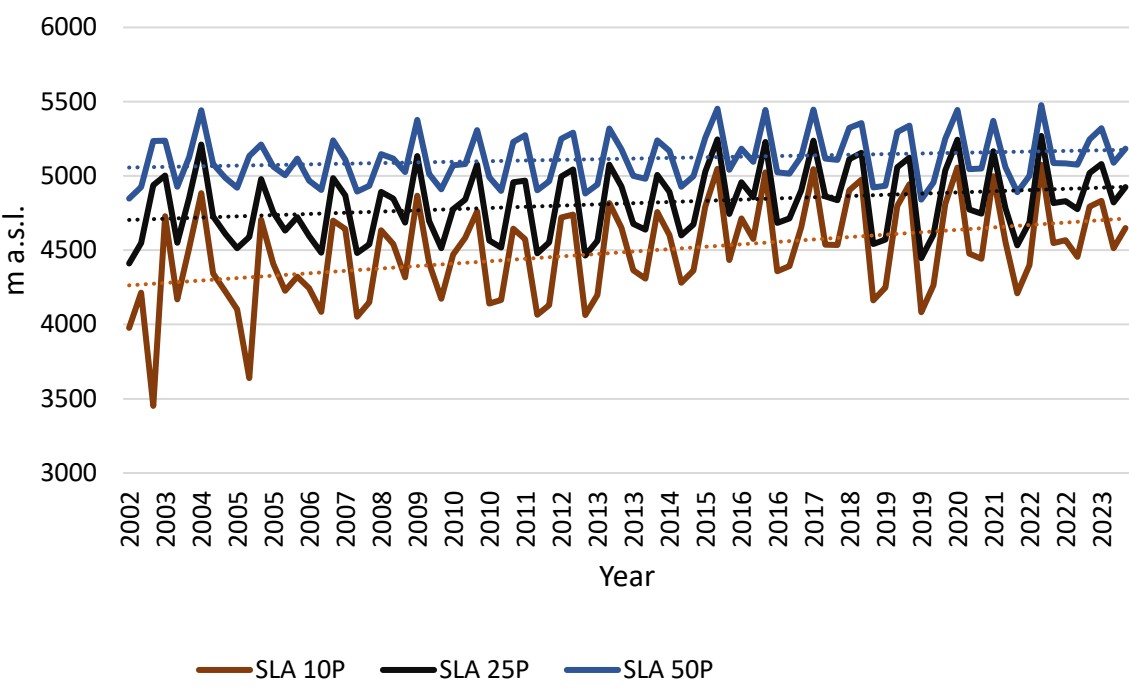


**Figure 14. Snowline shift using snow line of elevation of 10, 25 and 50 percentiles**

**5.0. Discussion**
This study **provides valuable** insights into the interactions between snow and ice **cover** in the
Upper Karnali Basin (UKB) and **the influencing climatic** and topographic factors. The results
reveal **significant t**rends and **variations** in snow cover, glacial retreat, **and snowline elevation**,
**consisten**t with broader climate change **patterns** observed in the Himalayan region. Below, we
discuss the key findings in relation to existing literature and their **implications** for water
resources, ecosystems, and local communities.
The study of the Upper Karnali Basin from 2002 to 2024 **offers** significant insights into the
relationship between snow cover area (SCA), temperature, and precipitation. The annual average
SCA is 872 km², with the highest snow cover occurring from January to March (1,528 ± 333
km²) and the lowest from July to September (169 ± 38.3 km²). The findings **indicate** a gradual
decline in snow cover across the Upper Karnali Basin (UKB) over this period, with an average
loss of approximately 3.99 km² **per year**.
There is a strong to moderate negative correlation between snow cover and temperature across
all seasons ($r = -0.59$ to $-0.77$, $p < 0.05$), signifying that higher temperatures **result** in reduced
snow cover. In contrast, precipitation **shows** a positive correlation with snow cover **during**
winter (January to March and October to December). The reduction in snow cover during the
winter months (January–March) **suggests** a potential shift in precipitation patterns, with more
precipitation falling as rain **rather than** snow. The winter and pre-monsoon snowpack in the
western Himalayas is heavily influenced by the Westerly wind system, which is a key source of
snowfall in the UKB (Syed et al., 2006; Dimri & Dash, 2012). Consequently, the decline in
winter snow cover may be **attributed not** only to temperature-induced changes in precipitation
but also to a possible weakening or **alteration** of the Westerlies, which **warrants** further
**investigation**. Such changes could lead to a decrease in overall moisture inflow (Yadav et al.,

545 2009).

This shift is temperature-dependent and, **consequently,** elevation-dependent, **leading to**
increased **snowmelt consistent** with global warming trends (Wester et al., 2019). During the
summer months (April to September), precipitation negatively correlates with snow cover, as it
**predominantly falls as** rain, further accelerating snowmelt. **Notably,** the period from July to
**September shows** a statistically significant decrease in snow cover (Sen's Slope = $-2.87$, $p <$
0.05), primarily driven by warmer temperatures and increased rainfall during **the summer**
**monsoon.**
Examining snow cover patterns in the UKB **sub-basins** reveals notable seasonal and spatial
variations. The Humla Karnali **sub-basin** has the largest average snow cover **from January to**
**March**, while downstream areas such as Tila and Kawari exhibit less snow **cover.**
The interannual variability in snow cover highlights the sensitivity of the snowpack to changing
temperature and precipitation patterns. This variability significantly **affects** water availability, as
the observed reduction in snow cover could exacerbate water scarcity during the dry season,
**impacting** millions who rely on snowmelt for irrigation, drinking water, and hydropower
generation (Immerzeel et al., 2020; Pritchard, 2019). The strong negative correlation in
**downstream Karnali** (r = -0.47, p < 0.05) further supports the declining trend in snow cover,
which threatens water availability and ecosystem services in the region (Wester et al., 2019).
The outcomes **highlight** the vulnerability of the UKB to climate change, as rising temperatures
and changing precipitation patterns result in reduced snow cover. **Implementing adaptive** water
management strategies is **essential to** mitigate the impacts on water resources and local
communities.
The findings on snow cover dynamics across elevation zones in the Upper Karnali Basin reveal
significant elevation-dependent patterns, reflecting the influence of temperature fluctuations and
global warming. At lower elevations (≤ 2000 m a.s.l.), snow cover exhibits a weak positive
correlation (0.12–0.43), likely **due to** occasional snowfall during brief cold spells and a transition
between rain and snow (Pendergrass, 2020). These zones experience high year-to-year variability
(CoV ~41–43%), making trends less reliable and warranting cautious interpretation. Similar
elevation-sensitive variability has also been reported in other Himalayan basins (Pepin et al.,

574 2015).


The transition from weak negative correlations **between snow cover, elevation**, and year above

2300 m a.s.l. to the strongest negative correlation at 6100–6200 m a.s.l. (r = -0.56) aligns with

evidence of elevation-dependent warming **(EDW)**. **In this phenomenon**, higher altitudes

experience accelerated warming, resulting in reduced snow accumulation and increased melt

rates. The sharp increase in mean snow cover above 5000 m a.s.l. **corresponds** to permanent

snow and glacier zones. However, the decline in inter-annual variability **(coefficient of**

**variation, CoV)** indicates a consistent reduction in snow cover, particularly at mid-to-high

elevations (3000–5000 m a.s.l.).

The nonlinear relationship between elevation and inter-annual snow cover variability (CoV) is

particularly insightful. **At elevations of 3,000 meters above sea level (a.s.l.)** or below, the CoV

reaches 41–43%, reflecting transitional zones where slight temperature fluctuations determine

the precipitation phase (rain versus snow). Above 3,000 **m a.s.l.,** the CoV decreases to 25–30%

as conditions remain persistently below freezing; however, the dominant driver shifts to

insolation and temperature-modulated melt rates. This observation aligns with Ren et al.'s (2023)

findings on Tibetan Plateau glaciers, where albedo feedbacks dominate mass balance above

**5,000** m a.s.l.

The strong negative correlation between land surface temperature and snow cover (Tau = -0.43

to -0.79) **underscores** the impact of rising temperatures on the snowpack. The most severe

declines occur between **3,000 and 5,000 meters above sea level (m a.s.l.)**, where warming

accelerates snowmelt and glacier retreat, threatening water availability for river flows,

agriculture, and hydropower (Immerzeel et al., 2020; Bolch et al., 2012).

Between 2000 and 2023, glacier basins in the Upper Karnali Basin experienced significant ice
and snow loss. The mean glacier area per basin declined from 119.04 to 100.47 **hectares,**
**representing** with an average loss of 18.6 hectares. While it occurred consistently across all
aspects, north-facing basins (N, NW) saw the largest total area decline. This trend, driven by
rising temperatures and reduced precipitation, results in a negative mass balance (Pepin et al.,
2022; Ren et al., 2024; Ye & Tian, 2022), threatening the persistence of glaciers and altering
critical water resources.
Snow cover trends in glacier basins reveal a consistent decline across all seasons. From January
to March, a majority (59%) of the 735 basins analyzed exhibit a statistically significant negative
correlation ($p < 0.05$), with 16.3% of all basins showing a substantial decline ($r < -0.44$). The
trend is even more **pronounced during** the post-monsoon and ablation seasons (October–
December). From July to September, 62% of basins ($n = 70$) show a significant negative
correlation, and in October–December, this figure rises to 91% ($n = 604$). This widespread
reduction in snow cover is linked to rising temperatures, which increase snowmelt rates and
reduce albedo, further accelerating glacier retreat (Dowson et al., 2020). These trends underscore
the vulnerability of the region's cryosphere to climate change, with profound implications for
water security and regional hydrology.
The seasonal snowline in the Upper Karnali Basin is rising **steadily** at rates of 5.6 m per year
(**10th percentile**), 2.91 m per year (**25th percentile**), and 1.54 m per year (50th percentile).
Although these rates are more conservative than many regional estimates, our findings align with
the broader Himalayan trend of snowline elevation. Recent studies **report** faster increases, such
as approximately 6.7–7.3 m per year in the Ganga–Brahmaputra basins (Dixit et al., 2024) and
roughly 8–14 m per year in several Nepalese catchments (Sasaki et al., 2024), while the
Langtang Basin shows a similar increase of about 2.2 m per year (Pradhananga et al., 2025). This
**pattern indicates** a consistent retreat of seasonal snow cover to higher elevations, **reducing** the
potential for snow accumulation to sustain glacier mass balance.

### 5.1. Feedback mechanisms and future projections

The correlation between temperature and snow cover ($\tau$ ranging from −0.43 to −0.79 across
different elevations) confirms the presence of a reinforcing snow–albedo feedback in the Upper
Karnali Basin (UKB). Increasing land surface temperatures reduce snow cover, lowering surface
albedo and **increasing the absorption of shortwave radiation**. This process causes localized
warming of **approximately** 0.8 to 1.2 °C, as estimated through Sen's slope analysis, further
promoting melting and accelerating the feedback loop. **Similar** snow–albedo feedback
mechanisms have been observed across the central and eastern Himalayas (**Brun et al., 2015**;
Bhattacharya et al., 2021; Salerno et al., 2023), underscoring the regional consistency of
cryospheric amplification.
In addition to snow cover analysis, glacier change data (Ghimire et al., 2025b) were integrated
with long-term temperature and precipitation records to assess cryospheric variability.
Relationships among temperature, snow cover, and glacier extent across elevation bands were
quantified using Kendall's $\tau$ and Sen's slope, providing estimates of warming trends and
snowline responsiveness. Future cryospheric conditions were simulated using a degree-day,
elevation-band glacio-hydrological model forced with bias-corrected CMIP6 (NEX-GDDP)
climate projections under the SSP1-2.6 and SSP2-4.5 scenarios, enabling projections of glacier
and snow cover evolution through 2100 (Ghimire et al., 2025b).
Above 5,000 m a.s.l., Sen's slope analysis indicates a mean warming rate of +0.064 °C per year,
comparable to the rates observed at mid-elevations (approximately +0.058 °C per year between
1,000 and 2,000 m). This elevation-dependent warming accelerates glacier thinning and shifts
the snow–rain boundary upward, thereby reducing accumulation periods and causing earlier melt
onset. Similar warming trends, with mean annual temperature increases of 0.05–0.07 °C per year
and glacier thinning rates of 0.3–1.0 m per year since 2000, have been documented in the central
Himalayas (Kääb et al., 2015; Bolch et al., 2019).
Under low-emission scenarios such as SSP1-2.6, high-altitude temperatures are projected to
increase by approximately 1 °C by 2100. Under the moderate SSP2-4.5 scenario, temperature
increases could reach 2 °C or more. Consequently, glacier areas are expected to decrease by 47–
69%, and snow-covered areas are projected to decline by 19–30% (Ghimire et al., 2025b). This
would transform the basin's hydrology from nival (snowmelt-dominated) to pluvial (rain-
dominated), increasing flood risks during monsoons and susceptibility to drought in dry seasons.
These projections **align** with other studies of Himalayan basins, which **predict** reductions in
glacier area of 40–60% by mid-century (Bhattacharya et al., 2021; Salerno et al., 2021; Hock et
al., 2019). **Similar** amplification mechanisms are also observed in the Andes and Alps, where
rapid glacier retreat and albedo-induced warming **reflect** trends seen in the Himalayas (Rabatel
et al., 2013; Vuille et al., 2018; Dussaillant et al., 2019; Beniston & Stoffel, 2014; Zemp et al.,

659     2019).


**6.0. Conclusions**

The study of snow and glacier cover dynamics in the Upper Karnali Basin from 2002 to 2024 reveals a persistent decline in snow cover, glacier area, and snowline elevation, driven by rising temperatures and **changes** in precipitation patterns.

The annual snow-covered area (SCA) has decreased by approximately 3.99 km² per year, with the most significant reductions **occurring** during the **July–September** monsoon period. This decline in snow cover is statistically correlated with **rising** temperatures, **highlighting** the impact of climate change on seasonal snow accumulation and melt cycles. **Variability** in winter snow cover suggests changes in snowfall patterns rather than a uniform decrease.

Notable seasonal and spatial differences in snow cover patterns are observed in the **sub-basins** of the UKB during **two periods**: January–March and October–December. The upstream sub-basins experience **more consistent** snowfall than the downstream basins. During October–December, snowfall is inconsistent **across** all basins, with particularly high variability in the China Karnali, **Tila, and downstream** Karnali basins.

Elevation-dependent trend analysis confirms that snow cover at lower elevations (<2000 m a.s.l.) exhibits high interannual variability, **whereas** higher elevations (>3000 m a.s.l.) show a significant long-term decline. The most pronounced reductions occur between 3000 and 5000 m a.s.l., where warming accelerates snowmelt and glacier retreat. The observed negative correlation between snow cover and rising temperatures confirms the climate-driven reduction in snowpack, exacerbating the risk of water shortages.

The study of glacier basins **reveals** widespread retreat, with the **average** glacier area **decreasing** from 119.05 hectares in 2000 to 100.47 hectares in 2023. Glacier retreat is most pronounced in

north-facing basins (N, NW, NE), where melting exceeds accumulation. The continuous decline
in snow cover across **these basins** indicates a **persistent** negative mass balance, **threatening the**
long-term **survival of the glaciers.**
Additionally, the snow line is gradually shifting upward, with the 10th, 25th, and 50th percentiles
rising by approximately 5.16, 2.91, and 1.54 meters per year**, respectively**. This **trend indicates**
a consistent loss of seasonal snow accumulation.
Given the current warming trends **(~0.0643°C per year** above 5000 m a.s.l.), the **Upper Karnali**
**Basin** (UKB) could experience a decline in glacier area by 47–69% and a reduction in snow-
covered area by 19–30%. This shift would transform the hydrology from snowmelt-dominated
(nival) to rainfall-dominated (pluvial), increasing the frequency of extreme weather events and
altering regional water security dynamics. These findings underscore the urgent need for
proactive water resource management, **enhanced** climate resilience strategies, and continuous
monitoring of cryospheric changes to mitigate future risks. Policymakers must prioritize
adaptation measures, such as improved water storage infrastructure and sustainable land-use
practices, to ensure long-term water security in the Upper Karnali Basin and beyond.
**Author contributions**
MG conceptualized the research, designed the methodology, conducted fieldwork, analyzed the
data, and drafted the manuscript. DS and RC assisted **with** proposal writing, research design,
fieldwork, and data analysis. AT, TPPS, KPS, SBG, and SD contributed to procuring remote
sensing and climate data. PB and SK were responsible for procuring and updating MODIS data.
WY reviewed the manuscript and provided feedback to enhance its quality. NT and JK assisted
**with** GIS analysis. All authors contributed to revising the manuscript and provided input before
submission.
**Competing Interests**
The authors declare that they have no conflict of interest.
**Data availability**
MODIS, Landsat, Sentinel, ERA5 **reanalysis** climate datasets, and NEX-GDDP data are
publicly available.

**Acknowledgments**


We express our gratitude to the Director of Tribhuvan University's Research Coordination and
Development Council (RCDC) for supporting the project titled "State and Dynamics of the
Cryosphere of the Upper Karnali Basin, Associated **Hazards,** and Implications for Water
Resources and Livelihood" (Project Code TU-NPAR-077/78-ERG-15). This paper is a product
of that project. We appreciate the Evaluation and Monitoring Committee of RCDC for their
insightful feedback and suggestions, which greatly enhanced the manuscript. We also thank the
University Grants Commission for providing research funding. **The authors sincerely**
acknowledge the Sichuan Science and Technology Program (2024YFHZ0248) for partial
support. **Additionally, we** acknowledge the contributions of Google Earth Engine, ERA5, ESRI,
and other open-access resources for providing satellite imagery and data.

**Financial support**


This research was funded by the University Grants Commission (UGC), Kathmandu, through
Tribhuvan University's Research Coordination and Development Council (RCDC) under the
National Policy Area Research program. **Partial funding was also provided by the Sichuan**
**Science and Technology Program (2024YFHZ0248).**

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
