# Peer review of "Dynamics of Snow and Glacier Cover in the Upper Karnali Basin, Nepal: An Analysis of"

_EGUsphere, 2025_

## Author Comment (AC2)

**Response to comments and suggestions from Reviewer 1**

GENERAL

A useful set of data on snow cover is presented for 22 years, and related to temperature, precipitation, elevation and time: data on glacier decline are also presented. A loss of cover is consistent but strongest at lower elevations. Inter-year variability is greatest in winter, and least for July-September. Temperature and precipitation is taken from reanalysis data, presumably based on sparse observations and with little control for higher elevations, so the results in section 4.2 should be accompanied by precautionary warnings.

A lot of clarification is needed, and the are some inconsistencies between text, Figures and Tables. It is not always clear what is being correlated with what. Perhaps 'trend' (temporal trend) should be used more often in the place of correlation, in some passages: e.g. 'negative trends with correlations over time of …'..

Some numbers have too many decimal places. Given that some error is inevitable, more rounding should be employed.

**Response:** Dear reviewer, thank you so much for your thorough and detailed evaluation of the manuscript. Your careful, line-by-line examination identified several errors, omissions, and unaddressed aspects within the text, figures, and tables. I have carefully reviewed my manuscript and incorporated almost all of your comments and suggestions.

In this study, land surface temperature (LST) data at a spatial resolution of 1 km were obtained for 204 locations from MODIS Terra (MYD11A1) and Aqua (MOD11A2) products, processed via the AppEEARS platform. Precipitation data were sourced from the ERA5-Land reanalysis dataset provided by ECMWF (Hersbach et al., 2020).

I acknowledge that the coarse spatial resolution of these datasets necessitates cautious analysis and interpretation. The manuscript does not currently address the variability in topography within the 1 km resolution of the MODIS-derived LST data or the precipitation data derived from reanalysis. Nonetheless, the importance of averaging over larger spatial units remains a significant consideration and should not be overlooked. Please review the comments from Reviewers 2 and 3, where I have made an effort to provide thorough responses.

Discussion is revised and made short without repetition.

SPECIFIC

Line 110  With such relief, surely precipitation must vary more than this?

**Response:** We have corrected. Precipitation varied from 250 to ~ 1900 mm annually.

136-143  What effect did the cloud removal have (in biasing coverage, both spatial and temporal)?

**Response:**

Relying on optical remote sensing data in the Himalayan mountain region presents a major limitation. Obtaining cloud-free imagery that consistently covers the entire area and time period is a significant challenge, especially during the pre-monsoon and monsoon seasons. As a result, snow is underestimated in cloud-covered zones, which can lead to potential inaccuracies in seasonal and spatial snow cover assessments.

To overcome this challenge, we combined high-temporal-resolution eight-day composite MODIS data with high-spatial-resolution Landsat imagery, which enabled effective monitoring and seasonal analysis. This integrated approach to a greater extent compensates for the limitations of individual datasets and supports consistent long-term cryospheric assessments in cloud-prone mountain regions like the Upper Karnali Basin.

Section 4.1 text implies a graph for annual cover is necessary: only the 4 seasons are illustrated..

**Response:** Annual average snow cover is included in the graph.

190 Sen's slope is not defined. It seems to be the gradient of the regression line over time, so why is attribution to 'Sen' needed?

**Respons**e: Sen's slope is defined, and its importance in analyzing the trend has been highlighted (see footnote of Table 1)

207 These Fig.2 graphs are initially puzzling in that Oct-Dec shows the steepest trend line but is insignificant, while July-Sept seems flatter but has the only significant trend. This seems to relate to the lower variability of July-Sept (SD 38 cf. 212-373, Table 1).

**Response:** The July–September period exhibits a gentler trend line compared to October–December; however, because of its much lower interannual variability (with a standard deviation of 38.3 km² compared to 212–373 km² for other seasons, as shown in Table 1), the trend remains statistically significant. The revised manuscript now includes an explanation clarifying how variability influences the determination of significance.

Comments: Why is the correlation positive below 2000 m (Fig.7) and below 2300 m, where the T is rising (Fig.8)? Are the data so limited below 2000 m that it should perhaps be dropped? Fig. 9 shows that warmer years have less snow cover, consistently across all elevations (although<2000 m is not shown).

**Response:** At lower elevations (≤2000 m a.s.l.), snow cover exhibits a weak positive correlation (0.12-0.43), likely caused by occasional snowfall during short cold spells and shift between rain and snow (Pendergrass, 2020). These zones experience high year-to-year variability (CoV ~41–43%), making trends less reliable, which should be interpreted with caution. Similar elevation-sensitive variability has also been reported in other Himalayan basins (Pepin et al., 2015).

DETAILED

88 'above'

**Response:** Above

90 'within Nepal'

**Response**: Corrected

107 and 150  Ghimire is not in References.

Response: Reference added  (Ghimire et al (2025).

118 This identifies 3 rivers , but not Kawari.  Also, the upper Himla is apparently labelled 'China Karnali' in Table 2, but that has not been specifically located..  There should be a closer relation between map and text (and Table).

**Response**: Map is corrected and missing information are included, text, and table  are updated and are matched with map. To clarify, Humla Karnali in Chinese territory has been labeled as Humla Karnali (China). Similarly, the downstream part of the Karnali has been labeled as Karnali (downstream).

132 delete 'then'

**Response**: Deleted

134  'sub-basin'

**Response**: Corrected to sub basin.

162  Why central?  not sub-glaciers.  Perhaps 'both glaciers and surrounding slopes'?  Is 'fed by' appropriate ?

**Response**: Corrected as "Glacier basins are areas that include trunk and sub-glaciers, along with surrounding slopes, which are nourished by moving ice and snow."

185 424?? Table 1 shows a July-Sept min of 169 and an annual min of 514.

**Response**:  corrected in the result and discussion with reference to Table 1. Measures of Mean, Max, and Min, and percentiles, and sen's slopes were included in the Table 1.

186  640.32 does not appear in the 25% row in Table 1.

**Response**: Corrected as in Table 1.

192 & 202 Unfortunately, Fig. 2 does not show annual averages.

**Response**: Annual average included in Figure 2

199-201  I am unsure what this sentence means and how it relates to Fig.2.  Also it needs a verb.

**Response:**  The unclear sentence has been revised as "Episodic snow coverage was observed in 2015, 2020, and 2022 (January–March), 2015 and 2019 (April–June), 2009, and 2021 (October–December), indicating anomalous years of high episodic heavy snowfall events"

204   Fig.2 The heading is unhelpful.  I suggest the more precise 'Annual and seasonal snow cover statistics (km2) with correlations of the trends, 2002-2024.'

**Response**: Corrected and caption of the figure has been revised.

204-5 Strange that Kendall's tau does not show a negative trend like all the other correlations.  Is tau appropriate here?

**Response:** In Table 1, Kendall's Tau complements Sen's Slope by indicating the statistical significance and direction of trends, especially emphasizing the significant decline in snow cover during July–September. Its inclusion enhances the robustness of trend analysis and supports key findings in the study. For April–June in Table 1, Kendall's Tau value is positive (Tau = +0.013, $p$ = 0.95). It indicates a statistically insignificant upward trend in snow cover during this season over the 2002–2024 period. This may be due to random year-to-year variation rather than a consistent long-term pattern.

210-211 Fig.3 does not show negative dominating: it is close to balance, with April-June (more negative) balancing Oct-Dec (more positive)

**Response:** Figure 3 has been enhanced by describing the number of locations showing positive and negative correlations. I have added another graph illustrating the annual temperature trend, which shows that approximately 70% of locations exhibit positive correlations. Although a few locations display statistically significant correlations ($p<0.1$), the overall trend remains positive.

[Figure]

[Figure]

Figure 3

212  Incorrect.  Fig.4  shows positive trends (probably insignificant) except for Jan-Mar.  Why 'June-July'?

**Response:** Corrected and revised the paragraph as

"Seasonal rainfall trends from 2000 to 2024 indicate weak to moderate increases across all seasons except winter (January–March), during which rainfall exhibits a slight downward trend ($R^2$= 0.0144) (Figure 4). Pre-monsoon (April–June) rainfall shows a slight upward trend ($R^2$ = 0.0119). All these

seasons display high variability, suggesting a limited impact on snow accumulation. Monsoon rainfall (July–September) demonstrates a more noticeable increase ($R^2 = 0.0975$), primarily contributing to rainfall rather than snowfall. Post-monsoon (October–November) precipitation remains low and stable. When combined with rising temperatures, these trends indicate a shift toward rainfall-dominated precipitation, reduced snowfall, and earlier snowmelt, contributing to declining snow cover and altered hydrological regimes".

215-217 This explanation of the 204 sampled should precede 210-211.

**Response:** The lines (215-217) describing the sources of temperature and precipitation data have been placed before 215-217.

219  should be '-0.59 to -0.77'

**Response**: Corrected

222-224  More concisely 'Precipitation and temperature are negatively correlated in winter (Oct-March) and positively in the summer (April-September) half-year'.

**Response:** I have replaced the previous lines with the above suggested lines.

225 Fig.3 How were the 19 correlations plotted here selected from the 206 or 204) ?  And perhaps the altitudes of these locations are important, explaining the wide variability / lack of spatial consistency?

**Response:** Figure 3 illustrates the temperature trends across various locations, which spatially range from negative to positive values. Only a few locations exhibit statistically significant trends in both directions, defined by a correlation coefficient exceeding ±0.4 and a p-value less than 0.05. In this figure, the altitudes of locations are not indicated. Given the spatial variability and inconsistency in the trends, we concur that altitude and topographic differences likely play a significant role. This aspect is further examined in Figures 7, 8, and 9, as well as Table 3, which analyze elevation-dependent warming patterns and correlations across different elevation bands. Additionally, a clarifying sentence has been added to the text to guide readers that elevation-dependent warming patterns and correlations across elevation bands will be discussed in the subsequent sections.

Fig. 4 would be improved if annual average values were connected by straight lines rather than curves: or if dots were used.

**Response:** Figure 4 is improved by connecting average values with straight lines. Similarly, curved line in others figures were also converted to straight line.

[Figure]

Figure 4

Fig.5 Larger numbers (on the coloured backgrounds) would clarify.

**Response:** Figure five improved as suggested

235  Presumably 'over the 22 years'.

**Response**: Included the number of years in the caption of Figure 5.

240 November?? What happened to December?

**Response:** Corrected

254 'the variability is strongest' is a duplication.

**Response:** Duplication removed

242-257  All this makes sense in terms of altitude: the lowest area (downstream) has the least and most variable snow cover, and a define decline with warming over the 22 years.

**Response:** The varying variability due to elevation difference and dependent warming  is Figure 7,8,and 9  and explained in the upcoming sections

269 &  285  State what snow cover is being correlated WITH – i. e. time?

**Response:** Explained by adding a sentence, "The seasonal correlation matrices (Figure 3,4, and 5) illustrate how Landsat-derived snow cover area is statistically correlated with corresponding seasonal temperature and precipitation, rather than with time itself. This helps identify the climatic controls on snow cover dynamics across seasons."

269  'in the lowest'

**Response**: Revised as 'lowest'

283 delete 'elevation'

**Response**: 'elevation' deleted

289  No:  Figure 8, not 5.

**Response:** Corrected as Figure 8

316 delete 'the'

**Response**: deleted

320-=321 What a truism!  Deleted the sentence

**Response:** Deleted, and I agree with the statement that was so obvious and therefore hardly worth mentioning.

334 delete '(able'

**Response**: deleted

335  Too many decimal places.  Drop '.163'  - of the order of a thousandth of a percent of the total area: surely spurious accuracy.

**Response:** Corrected, reduced to one decimal place.

337  Drop ', indicating a relative reduction in glacier coverage'  - another unnecessary truism.

**Response:** Deleted the obvious part of the sentence.

342  Yes, but S shows the largest absolute loss.  You might also consider the relative (%) loss for each direction class.

**Response:** Relative (%) losses for each slope direction class have been included in the text.

343 Delete 'Northeast (NE),' which is repeated.

**Response:** Deleted the repeated ones

Fig.10: The order is illogical; these should be in rank order, e.g., NW, N, NE, E, SE, S, SW, W.

**Response:**  Slope directions of the glacier has been put in appropriate order in fugure.

358  "May , June & July" straddles two of the seasons in the Figures.

**Response:** Corrected as May–July

361  delete '(n='

**Response**: Deleted

361  "84%" is not apparent in any part of Fig.11.

**Response:** The whole Figure is corrected and updated  and the text has been revised.

[Figure]

363  Fig. 11  Does the orange line relate to all basins, rather than just the selection whose IDs are given?

**Response:** Due to cloud cover, not all glacier basins for different seasons were analyzed. Therefore, the orange line does not represent all basins, but only cloud-free glacier basins in a given season.

369 'in the remaining basins'

**Response:** Corrected

415  What does "although this precipitation does not appear to facilitate snow accumulation" mean? Mean?  Where is the evidence?

**Response:** A positive correlation between precipitation and snow accumulation does not necessarily imply a direct causal relationship whereby precipitation contributes to snow accumulation. Although these phenomena may coincide temporally, precipitation may occur in forms other than snow. In the

Upper Karnali Basin, particularly during the winter months, warmer winter temperatures associated with climate change can result in precipitation falling as rain rather than snow. Regional and global observations indicating that warming trends increase the proportion of rainfall even during seasons traditionally characterized by snowfall (e.g., Wester et al., 2019; Kraaijenbrink et al., 2021).

417  "rain instead of snow" is temperature-dependent and thus elevation-dependent.

**Response:** The line has been revised and has incorporated the above.

431 ?  exhibits …   'less'?

**Response:** Corrected as "exhibits less snow cover".

432-437  duplicates 423-428.  Poor editing!

**Response:** It was a mistake; the duplicate paragraph has been removed.

454 Yet Fig. 8 and line 293 suggest reduced warming high up.

**Response:**

460  'inter-annual snow cover variability '

**Response**: Included 'inter' in the sentence

461 3700 m ?  from Fig.8.

**Response:** Corrected as above 3000 m

462 4100 m?   "

**Response:** Corrected

473 'reveal'

**Response:** Corrected

474-475 Too many decimal places.

**Response:** Decimal place reduce to one or two.

528—529  Decimal places !

**Response:** Decimal place reduced.

571  delete one 2014

**Response:** deleted

589  give authors

**Response**: Reference deleted

595 delete "(last …."

**Response:** Deleted from the reference

**Citation**: https://doi.org/10.5194/egusphere-2025-1303-RC1

**Response:** Will be cited**

---

## Author Response (AR1)

**Acknowledgment to Editor and Reviewers**

We are happy to resubmit our revised manuscript **(egusphere-2025-1303)** entitled "**Dynamics of Snow and Glacier Cover in the Upper Karnali Basin, Nepal: An Analysis of Its Relationship with Climatic and Topographic Parameters.**"

We sincerely thank the Editor and Reviewers for their thorough assessment of our manuscript and for their valuable, insightful comments. Their detailed feedback has significantly enhanced the scientific rigor, clarity, and overall presentation of our work. We appreciate the time and effort they dedicated to helping us improve this study, and we are grateful for the opportunity to revise and resubmit our manuscript.

**Response to Reviewer 1**

GENERAL

A useful set of data on snow cover is presented for 22 years, and related to temperature, precipitation, elevation and time: data on glacier decline are also presented. A loss of cover is consistent but strongest at lower elevations. Inter-year variability is greatest in winter, and least for July-September. Temperature and precipitation is taken from reanalysis data, presumably based on sparse observations and with little control for higher elevations, so the results in section 4.2 should be accompanied by precautionary warnings.

A lot of clarification is needed, and the are some inconsistencies between text, Figures and Tables. It is not always clear what is being correlated with what. Perhaps 'trend' (temporal trend) should be used more often in the place of correlation, in some passages: e.g. 'negative trends with correlations over time of …'..

Some numbers have too many decimal places. Given that some error is inevitable, more rounding should be employed.

**Response:** Dear reviewer, thank you so much for your thorough and detailed evaluation of the manuscript. Your careful, line-by-line examination identified several errors, omissions, and unaddressed aspects within the text, figures, and tables. I have carefully reviewed my manuscript and incorporated almost all of your comments and suggestions. See the revised version of manuscript.

In this study, land surface temperature (LST) data at a spatial resolution of 1 km were obtained for 204 locations from MODIS Terra (MYD11A1) and Aqua (MOD11A2) products, processed via the AppEEARS platform. Precipitation data were sourced from the ERA5-Land reanalysis dataset provided by ECMWF (Hersbach et al., 2020). **See lines 181-196; 235-245.**

I acknowledge that the coarse spatial resolution of these datasets necessitates cautious analysis and interpretation. The manuscript does not currently address the variability in topography within the 1 km resolution of the MODIS-derived LST data or the precipitation data derived from reanalysis. Nonetheless, the importance of averaging over larger spatial units remains a significant consideration and should not be overlooked. Please review the comments from Reviewers 2 and 3, where I have made an effort to provide thorough responses. **See line 114**

Discussion is revised and made short without repetition.

SPECIFIC

Line 110  With such relief, surely precipitation must vary more than this?

**Response:** We have corrected. Precipitation varied from 250 to ~ 1900 mm annually. **See lines 500-625.**

136-143  What effect did the cloud removal have (in biasing coverage, both spatial and temporal)?

**Response:**

Relying on optical remote sensing data in the Himalayan mountain region presents a major limitation. Obtaining cloud-free imagery that consistently covers the entire area and time period is a significant challenge, especially during the pre-monsoon and monsoon seasons. As a result, snow is underestimated in cloud-covered zones, which can lead to potential inaccuracies in seasonal and spatial snow cover assessments.

To overcome this challenge, we combined high-temporal-resolution eight-day composite MODIS data with high-spatial-resolution Landsat imagery, which enabled effective monitoring and seasonal analysis. This integrated approach to a greater extent compensates for the limitations of individual

datasets and supports consistent long-term cryospheric assessments in cloud-prone mountain regions like the Upper Karnali Basin. **See lines 152-158; 230-234**

Section 4.1 text implies a graph for annual cover is necessary: only the 4 seasons are illustrated..

**Response:** Annual average snow cover is included in the graph. **See page 16, line 279.**

190 Sen's slope is not defined. It seems to be the gradient of the regression line over time, so why is attribution to 'Sen' needed?

**Respons**e: Sen's slope is defined, and its importance in analyzing the trend has been highlighted (see footnote of Table 1) **See Page 15, lines 281-284.**

207 These Fig.2 graphs are initially puzzling in that Oct-Dec shows the steepest trend line but is insignificant, while July-Sept seems flatter but has the only significant trend. This seems to relate to the lower variability of July-Sept (SD 38 cf. 212-373, Table 1).

**Response:** The July–September period exhibits a gentler trend line compared to October–December; however, because of its much lower interannual variability (with a standard deviation of 38.3 km² compared to 212–373 km² for other seasons, as shown in Table 1), the trend remains statistically significant. The revised manuscript now includes an explanation clarifying how variability influences the determination of significance**. See Page 13, lines 259-262**

Comments: Why is the correlation positive below 2000 m (Fig.7) and below 2300 m, where the T is rising (Fig.8)? Are the data so limited below 2000 m that it should perhaps be dropped? Fig. 9 shows that warmer years have less snow cover, consistently across all elevations (although<2000 m is not shown).

**Response:** At lower elevations (≤2000 m a.s.l.), snow cover exhibits a weak positive correlation (0.12-0.43), likely caused by occasional snowfall during short cold spells and shift between rain and snow (Pendergrass, 2020). These zones experience high year-to-year variability (CoV ~41–43%), making trends less reliable, which should be interpreted with caution. Similar elevation-sensitive variability has also been reported in other Himalayan basins (Pepin et al., 2015).**See Page 23, lines 267-272**

DETAILED

88 'above'

**Response:** Above   **See page 5, line 103**

90 'within Nepal'

**Response**: The whole text in the introduction has been revised following Review 4's comment. **Page 3-5, lines 41 to 99.**

107 and 150  Ghimire is not in References.

Response: Reference added  Ghimire et. al., (2025a and b). **See page 48-49, lines 778-787.**

118 This identifies 3 rivers , but not Kawari.  Also, the upper Himla is apparently labelled 'China Karnali' in Table 2, but that has not been specifically located..  There should be a closer relation between map and text (and Table).

**Response**: Map is corrected and missing information are included, text, and table  are updated and are matched with map. To clarify, Humla Karnali in Chinese territory has been labeled as Humla Karnali (China). Similarly, the downstream part of the Karnali has been labeled as Karnali (downstream). **See page 7, lines 128.**

132 delete 'then'

**Response**: Whole paragraph rephrased**. See page 8, lines 143-151**

134  'sub-basin'

**Response**: Corrected to sub basin. **See page 8, lines 150.**

162  Why central?  not sub-glaciers.  Perhaps 'both glaciers and surrounding slopes'?  Is 'fed by' appropriate ?

**Response**: Corrected as "Glacier basins are areas that include trunk and sub-glaciers, along with surrounding slopes, which are nourished by moving ice and snow". **See page 11, lines 199-202**

185 424?? Table 1 shows a July-Sept min of 169 and an annual min of 514.

**Response**: corrected in the result and discussion with reference to Table 1. Measures of Mean, Max, and Min, and percentiles, and sen's slopes were included in the Table 1. **See page 15, lines 275-276.**

186 640.32 does not appear in the 25 % row in Table 1.

**Response**: Corrected as in Table 1. **See page 15, lines 275-276.**

192 & 202 Unfortunately, Fig. 2 does not show annual averages.

**Response**: Annual average included in Figure 2. **See page 16, 279.**

199-201 I am unsure what this sentence means and how it relates to Fig.2. Also it needs a verb.

**Response:** The unclear sentence has been revised as "Episodic snow coverage was observed in 2015, 2020, and 2022 (January–March), 2015 and 2019 (April–June), 2009, and 2021 (October–December), indicating anomolous years of high episodic heavy snowfall events".**See page 13-14, lines 268-271.**

204 Fig.2 The heading is unhelpful. I suggest the more precise 'Annual and seasonal snow cover statistics (km2) with correlations of the trends, 2002-2024.'

**Response**: Corrected and caption of the figure has been revised. **See page 16, lines 281-284**

204-5 Strange that Kendall's tau does not show a negative trend like all the other correlations. Is tau appropriate here?

**Response:** In Table 1, Kendall's Tau complements Sen's Slope by indicating the statistical significance and direction of trends, especially emphasizing the significant decline in snow cover during July–September. Its inclusion enhances the robustness of trend analysis and supports key findings in the study. For April–June in Table 1, Kendall's Tau value is positive (Tau = +0.01, p =

0.95). It indicates a statistically insignificant upward trend in snow cover during this season over the 2002–2024 period. This may be due to random year-to-year variation rather than a consistent long-term pattern. **See page 15, lines 276**

210-211 Fig.3 does not show negative dominating: it is close to balance, with April-June (more negative) balancing Oct-Dec (more positive).

**Response:** Figure 3 has been enhanced by describing the number of locations showing positive and negative correlations. I have added another graph illustrating the annual temperature trend, which shows that approximately 70% of locations exhibit positive correlations. Although a few locations display statistically significant correlations (p<0.1), the overall trend remains positive. **See page 18, lines 318-319.**

[Figure]

[Figure]

Figure 3

212  Incorrect.  Fig.4  shows positive trends (probably insignificant) except for Jan-Mar.  Why 'June-July'?

**Response:** Corrected and revised the paragraph as

Seasonal rainfall trends from 2000 to 2024 indicate weak to moderate increases across all seasons, with the exception of winter (January–March), which exhibits a slight downward trend (R²= 0.0144) (Figure 4). Pre-monsoon (April–June) rainfall shows a slight upward trend (R² = 0.0119). All these seasons display high variability, suggesting a limited impact on snow accumulation. Monsoon rainfall (July–September) demonstrates a more noticeable increase (R² = 0.0975), primarily contributing to rainfall rather than snowfall. Post-monsoon (October–December) precipitation remains low and stable. When combined with rising temperatures, these trends indicate a shift toward rainfall-dominated precipitation, reduced snowfall, and earlier snowmelt, contributing to declining snow cover and altered hydrological regimes. **See page 17, lines 302-310**

215-217 This explanation of the 204 sampled should precede 210-211.

**Response:** The lines (215-217) describing the sources of temperature and precipitation data have been placed before 215-217. **See page 16, lines 286-289.**

219  should be '-0.59 to -0.77'

**Response**: Corrected. **See page 17, lines 311-312**

222-224  More concisely 'Precipitation and temperature are negatively correlated in winter (Oct-March) and positively in the summer (April-September) half-year'.

**Response:** I have replaced the previous lines with the above suggested lines. **See page 18, lines 315-316**

225 Fig.3 How were the 19 correlations plotted here selected from the 206 or 204) ?  And perhaps the altitudes of these locations are important, explaining the wide variability/lack of spatial consistency?

**Response:** Figure 3 illustrates the temperature trends across various locations, which spatially range from negative to positive values. Only a few locations exhibit statistically significant trends in both directions, defined by a correlation coefficient exceeding ±0.4 and a p-value less than 0.05. In this figure, the altitudes of locations are not indicated. Given the spatial variability and inconsistency in the trends, we concur that altitude and topographic differences likely play a significant role. This aspect is further examined in Figures 7, 8, and 9, as well as Table 3, which analyze elevation-dependent warming patterns and correlations across different elevation bands. Additionally, a clarifying sentence has been added to the text to guide readers that elevation-dependent warming patterns and correlations across elevation bands will be discussed in the subsequent sections. **See page 24, section  4.4. and lines 367-383.**

Fig. 4 would be improved if annual average values were connected by straight lines rather than curves: or if dots were used. **See page 24, section  4.4. and lines 367-383.**

**Response:** Figure 4 and 6 is improved by connecting average values with straight lines. Similarly, curved line in others figures were also converted to straight line. **See page 19 and 23.**

And 23

[Figure]

Figure 4

Fig.5 Larger numbers (on the coloured backgrounds) would clarify.

**Response:** Figure five improved as suggested. **See page 20, line 331.**

235  Presumably 'over the 22 years'.

**Response**: Included the number of years in the caption of Figure 5. **See page 20, line 331.**

240 November?? What happened to December?

**Response:** Corrected. **See page 20, line 337.**

254 'the variability is strongest' is a duplication.

**Response:** Duplication removed. **See page 21, lines 344-345**

242-257  All this makes sense in terms of altitude: the lowest area (downstream) has the least and most variable snow cover, and a define decline with warming over the 22 years.

**Response:** The varying variability due to elevation differences and dependent warming is described in Figures 7, 8, and 9 and explained in the upcoming sections. **See page 25, 26, and 27.**

269 &  285  State what snow cover is being correlated WITH – i. e. time?

**Response:** The snow cover in various elevation bands is correlated with time. **See pages 23, and 25 and lines 369 and 386.**

269 'in the lowest' Correct.

**Response**: Revised as 'lowest'. **See page 23 and line 369.**

283 delete 'elevation'

**Response**: 'elevation' deleted. **See page 24, line 384.**

289 No: Figure 8, not 5.

**Response:** Corrected as Figure 8. **See page 25, line 391.**

316 delete 'the'

**Response**: deleted. **See page 28, line 418.**

320-=321 What a truism! Deleted the sentence

**Response:** Deleted, and I agree with the statement that was so obvious and therefore hardly worth mentioning. **See page 29, lines 421-424.**

334 delete '(able'

**Response**: deleted. **See page 29, lines 434.**

335 Too many decimal places. Drop '.163' - of the order of a thousandth of a percent of the total area: surely spurious accuracy.

**Response:** Corrected, reduced to one decimal place. **See page 29, lines 434-435.**

337 Drop ', indicating a relative reduction in glacier coverage' - another unnecessary truism.

**Response:** Deleted the obvious part of the sentence. **See page 29, lines 437.**

342  Yes, but S shows the largest absolute loss.  You might also consider the relative (%) loss for each direction class.

**Response:** Relative (%) losses for each slope direction class have been included in the text. **See page 30, lines 444.**

343 Delete 'Northeast (NE),' which is repeated.

**Response:** Deleted the repeated ones. **See page 30, lines 443.**

Fig.10: The order is illogical; these should be in rank order, e.g., NW, N, NE, E, SE, S, SW, W.

**Response:**  Slope directions of the glacier has been put in appropriate order in Figure 10. **See page 30, lines 446.**

358  "May , June & July" straddles two of the seasons in the Figures.

**Response:** Corrected as May–July. **See page 31, lines 456.**

361  delete '(n='

**Response**: Deleted

361  "84%" is not apparent in any part of Fig.11.

**Response:** The whole Figure is corrected and updated  and the text has been revised. **See page 31, lines 456-462**

[Figure]

363 Fig. 11 Does the orange line relate to all basins, rather than just the selection whose IDs are given?

**Response:** Due to cloud cover, not all glacier basins for different seasons were analyzed. Therefore, the orange line does not represent all basins, but only cloud-free glacier basins in a given season. **See page 31, lines 462.**

369 'in the remaining basins'

**Response:** Corrected. **See page 32, lines 469.**

415 What does "although this precipitation does not appear to facilitate snow accumulation" mean? Mean? Where is the evidence?

**Response:** A positive correlation between precipitation and snow accumulation does not necessarily imply a direct causal relationship whereby precipitation contributes to snow accumulation. Although these phenomena may coincide temporally, precipitation may occur in forms other than snow. In the Upper Karnali Basin, particularly during winter months, warmer winter temperatures associated with

climate change can lead to precipitation falling as rain rather than snow. Regional and global observations indicate that warming trends increase rainfall even during seasons traditionally characterized by snowfall (e.g., Wester et al., 2019; Kraaijenbrink et al., 2021). **See page 36, lines 514-525.**

417 "rain instead of snow" is temperature-dependent and thus elevation-dependent.

**Response:** The line has been revised and has incorporated the above. **See page 36, lines 519-520.**

431 ? exhibits … 'less'?

**Response:** Corrected as "exhibits less snow cover". **See page 36, line 528.**

432-437 duplicates 423-428. Poor editing!

**Response:** It was a mistake; the duplicate paragraph has been removed. **See page 36, lines 529-533.**

454 Yet Fig. 8 and line 293 suggest reduced warming high up.

**Response:** Rephrased and explained**. See page 37 and 38, lines 548-566**

460 'inter-annual snow cover variability '

**Response**: Included 'inter' in the sentence**. See page 38, line 555.**

461 3700 m ? from Fig.8.

**Response:** Corrected as above 3000 m. **See page 38, line 555.**

462 4100 m? "

**Response:** Corrected. **See page 38, line 555**

473 'reveal'

**Response:** Corrected. **page 38, line 573**

474-475 Too many decimal places.

**Response:** Decimal place reduce to one or two. **page 38, lines 568-569**

528—529 Decimal places !

**Response:** Decimal place reduced. **page 42, line 647**

571 delete one 2014.

**Response:** All references have been checked and formatted according to The Cryosphere format. **See pages 44-55.**

589 give authors

**Response**: All references have been checked and formatted. **See pages 44-55.**

595 delete "(last …."

**Response:** All references have been checked and formatted. **See pages 44-55.**

**Citation**: https://doi.org/10.5194/egusphere-2025-1303-RC1

**Response: C**ited.

**References Cited in this Response**

Hersbach, H., Bell, B., Berrisford, P., Hirahara, S., Horányi, A., Muñoz-Sabater, J., Nicolas, J., Peubey, C., Radu, R., Schepers, D., Simmons, A., Soci, C., Abdalla, S., Abellan, X., Balsamo, G., Bechtold, P., Biavati, G., Bidlot, J., Bonavita, M., De Chiara, G., Dahlgren, P., Dee, D., Diamantakis, M., Dragani, R., Flemming, J., Forbes, R., Fuentes, M., Geer, A., Haimberger, L., Healy, S., Hogan, R. J., Hólm, E., Janisková, M., Keeley, S., Laloyaux, P., Lopez, P., Lupu, C., Radnoti, G., de Rosnay, P., Rozum, I., Vamborg, F., Villaume, S., and Thépaut, J.-N.: The ERA5 global reanalysis, Q. J. Roy. Meteor. Soc., 146, 1999–2049, https://doi.org/10.1002/qj.3803, 2020.

Kraaijenbrink, P. D. A., Stigter, E. E., Yao, T., and Immerzeel, W. W.: Climate change decisive for Asia's snow meltwater supply, Nat. Clim. Change, 11, 591–597, https://doi.org/10.1038/s41558-021-01074-x, 2021.

Pendergrass, A. G.: The global-mean precipitation response to CO2-induced warming in CMIP6 models, Geophys. Res. Lett., 47, e2020GL089964, https://doi.org/10.1029/2020GL089964, 2020.

Pepin, N., Bradley, R. S., Diaz, H. F., Baraer, M., Caceres, E. B., Forsythe, N., Fowler, H., Greenwood, G., Hashmi, M. Z., Liu, X. D., Miller, J. R., Ning, L., Ohmura, A., Palazzi, E., Rangwala, I., Schöner, W., Severskiy, I., Shahgedanova, M., Wang, M. B., Williamson, S. N., and Yang, D. Q.: Elevation-dependent warming in mountain regions of the world, Nat. Clim. Change, 5, 424–430, https://doi.org/10.1038/nclimate2563, 2015.

Wester, P., Mishra, A., Mukherji, A., and Shrestha, A. B. (Eds.): The Hindu Kush Himalaya Assessment: Mountains, Climate Change, Sustainability and People, Springer International Publishing, Cham, https://doi.org/10.1007/978-3-319-92288-1, 2019.

**Response to Reviewer 2**

The manuscript "Dynamics of snow and glacier cover in the Upper Karnali Basin, Nepal: An analysis of its relationship with climatic and topographic parameters" presents a significant and timely study of snow and glacier changes in the Upper Karnali Basin, Nepal. The authors have successfully analyzed snow covered area(SCA) change during 2002 and 2023, and the relationship between SCA and climate change. This work highlights SCA, as a critical, yet hitherto under-researched, component of the region's water resource, especially when compared to glaciers. The study's findings have important implications for understanding regional hydrology, assessing regional water security. The manuscript is well-written and the conclusions are well-supported by the data. I recommend acceptance of this manuscript after minor revisions.

**Response:** Thanks for your valuable and insightful comments and suggestions. In the manuscript, while addressing the reviewers' comments, I came across several mistakes and inconsistencies, and identified issues in text, tables, figures, and references. I have attempted to address and incorporate the comments and suggestions with sincerity and care.

**General comments**

1. Data sources and methods should be given much more detail. Authors mentioned that MOD10A1 was used to analyze the SCA, and also pointed out that cloud-masked snow cover data was classified into four seasons and calculated using a threshold-based binary mask. But Authors must analyze the uncertainty of SCA due to cloud-masked.

    **Response:** To address the issue of cloud-contaminated pixels in daily MODIS snow cover data, we also utilized the MOD10A2 8-day composite product, which applies a maximum snow extent algorithm across an 8-day window. This approach significantly reduces cloud-induced gaps by retaining the clearest observation for each pixel, thereby increasing spatial coverage and improving the reliability of seasonal snow estimates in cloudy months (Parajka and Blöschl, 2008). While it loses daily temporal resolution, the 8-day composite effectively smooths out short-lived cloud effects, offering a more stable dataset for trend analysis. **See page 8, 9, 12; lines 152-158, 170-173, 228-234.**

2. There are two SCA, one is derived from the resolution of MODIS products (500m). Another is derived from Landsat (30m). However, authors did not describe how to combine both SCAs. In addition, The Higher resolution from Landsat could be used to evaluate the uncertainty of MODIS products. But I do not see there is uncertainty evaluation.

**Response:** I agree that the disparity in spatial resolution between MODIS (500 m) and Landsat 8 (30 m) can result in mixed pixels within MODIS data, especially in regions characterized by patchy snow cover. The uncertainty associated with MODIS-derived snow cover area (SCA) can be assessed by employing higher-resolution Landsat 8 imagery as a benchmark. The finer 30 m resolution of Landsat 8 provides more precise delineation of snow cover, particularly in heterogeneous landscapes, rendering it an appropriate reference for evaluating the coarser 500 m MODIS product. **See Page 9 and 12, lines 170-173; 230-234. Supplement, Appendix B.**

*Sampling and Data Preparation*

To facilitate comparison at a consistent spatial scale, we resampled Landsat 8 SCA maps to a 500 m resolution to match MODIS grid cells. Each Landsat scene covers an area of $185 \times 180$ km. From each resampled scene, 10% of the total pixels (approximately 13,320 pixels) were randomly selected as vector points using ArcGIS. These points represent snow or non-snow classes (including cloud-covered areas).

The layer of random points was overlaid on the corresponding MODIS SCE maps, and the MODIS snow classification values were extracted to the attribute table of each point. This created a composite attribute table containing snow/non-snow classifications from both Landsat 8 and MODIS for each sampled point.

*Accuracy Assessment and Confusion Matrix*

Accuracy assessment was conducted using a confusion matrix (Table 1) to evaluate the agreement between the two products. The analysis focused on six scenes selected based on a <7% cloud cover threshold to minimize misclassification due to cloud obstruction. Two scenes were from the Upper Karnali Basin region, and one was from eastern Nepal.

**Table B1.** Confusion matrix showing the matching of pixels of Snow cover extent derived from MODIS and Landsat 8 /Sentinel 2 as a measure of accuracy the processed MODIS snow product (Example).

|  |  | MODIS | | | |
| --- | --- | --- | --- | --- | --- |
|  |  | **Non snow** | **Snow** | **Total** | **User's accuracy** |
|  | **Non snow** | 8950 | 2472 | 11422 | **78.35** |
| *Landsat (30 m) resample to 500 m* | **Snow** | 144 | 4744 | 4888 | **97.05** |
|  | **Total** | 9094 | 7216 | 16310 |  |
|  | **Producer's accuracy** | **98.4** | **65.7** |  |  |

| Overall accuracy | 83.97 |
|---|---|
| Bias | 0.93 |

*Results*

The comparison of MODIS and Landsat 8 SCA classification yielded overall accuracies (OA) ranging from 77.5% to 94.9% across the six evaluated scenes (Table 2). These findings align with previous validation studies and demonstrate that while MODIS provides a reliable estimate of snow cover at a broader scale, resolution-induced uncertainty exists and can be quantified effectively using higher-resolution data such as Landsat 8.

**Table B2.** Description of Landsat 8 surface reflectance data (Bands 3, 6, 5, and 9) used for validating daily MODIS Snow Cover Extent (SCE), including acquisition date, cloud cover percentage, overall classification accuracy, and bias. The last column indicates the region of the scene (MW = Mid-Western, FW = Far-Western, E = Eastern Nepal).

| S.N | Date | Cloud cover % | Overall accuracy | Bias | Region of Nepal |
|---|---|---|---|---|---|
| 1 | 03/08/2020 | 6.52 | 83.96 | 0.93 | MW |
| 2 | 03/27/2021 | 4.45 | 94.89 | 0.86 | MW |
| 3 | 03/02/2021 | 5.2 | 86.21 | 0.88 | FW |
| 4 | 12/31/2021 | 2.36 | 77.59 | 0.99 | FW |
| 5 | 23/04/2021 | 6.9 | 97.62 | 0.94 | E |
| 6 | 21/12/2019 | 5.29 | 97.22 | 0.78 | E |

A comparative analysis was conducted between snow cover extent derived from MODIS data (spatial resolution of 500 meters) and that obtained from Landsat-8 imagery (spatial resolution of 30 meters) on a sub-scene basis to evaluate the accuracy of the datasets (Table 3) The observed discrepancies in snow cover extent ranged from approximately 1.3 to 1.6 times. The MODIS data exhibited spatial overestimation of snow cover extent, attributable to its coarser spatial resolution. Therefore, it is essential to recognize the magnitude of data discrepancies arising from differences in spatial resolution when interpreting snow cover information

**Table B3**. Comparison of snow-covered pixel counts and areas derived from MODIS (500 m resolution) and Landsat 8 (30 m resolution) for selected dates. The exaggeration factor represents the ratio of the MODIS-derived snow-covered area to that of Landsat 8, highlighting the potential overestimation caused by the coarser spatial resolution.

| Date of scene | | Unit | MODIS 500 m | Landsat (30 m) | Exaggeration factor |
|---|---|---|---|---|---|
| 3/8/2020 | MW | Count | 22878 | 4758571 | |
| | | Area (ha) | 432200 | 571950 | 1.3 |
| 3/27/2021 | MW | Count | 6587 | 1342895 | |
| | | Area (ha) | 171425 | 120860 | 1.4 |
| 2/3/2021 | FW | Count | 4736 | 815756 | |
| | | Area (ha) | 118400 | 73418 | 1.6 |
| 12/31/2021 | FW | Count | 12042 | 2037280 | |
| | | Area (ha) | 301075 | 183355 | 1.6 |

3. GFor the land surface temperature (LST), there is difference between glacier surface temperature and other surface cover (Wu et al., 2015). I'm skeptical of the existing results.

**Response:** In this investigation, snow and glacier cover are considered collectively as a unified cryospheric component due to their analogous functional roles. Moderate Resolution Imaging Spectro radiometer (MODIS) land surface temperature (LST) data at a spatial resolution of 1 km were utilized to examine temperature trends across these combined areas, rather than isolating glacier-specific thermal measurements. Although appropriate for analyses at the basin scale, MODIS data lack the spatial granularity necessary to resolve fine-scale thermal heterogeneity on glacier surfaces. Wu et al. (2015) demonstrated that glacier surface temperatures exhibit variability influenced by factors such as albedo, shading, and surface roughness, employing a split-window algorithm with Landsat ETM+ imagery to attain enhanced accuracy (root mean square error approximately 1.2°C). Consequently, MODIS LST data are interpreted herein with respect to seasonal and elevation patterns, while recognizing their limitations for detailed assessments focused exclusively on glacier-specific temperature dynamics.

This clarification has been incorporated into the discussion section to address the reviewer's concern. And the prescribed references has been incorporated. **See Page 7 and 12; lines 132-135, 235-246.**

**Specific comments**

Line45 therby-thereby

**Response:** Corrected as 'thereby'. **See page 3, line 49**

Line 194 P=0.00??

**Response:** Corrected to 0.001. **See page 13, line 263**

Figure 6 have to mark the sub-figure as a,b,c,d, those sub-figures are also be explained in title. The same as Figure 9,11

**Response:** All sub-figures were marked as A,B,C… **See Figures 2-6 and 11 on pages 10, 11, 19, 20, 23, and 31 respectively.**

Figure 8, The temperatures for different elevation bins were shown in Figure 8. It is very nice to show the temperature rate along with the elevation. However, I do not know where the data of temperature come?? Is it LST or ERA5?

**Response:** It is MODIS derived LST. Incorporated in Figure 8**. See page 26, lines 401 and 402.**

Line 316 Snow cover the trend in Glacier Basins-> The snow cover trend in Glacier Basins

**Response:** Corrected. **See page 28, line 420.**

 Figure 11 January-march_ "delete _"

**Response:** Deleted "_"**See page 31, line 464.**

Line 431 exhibits what???

**Response:** Corrected**. …**exhibits less snow cover. **See page 36, line 530**

**References Cited in this Response**

Parajka, J. and Blöschl, G.: Spatio-temporal combination of MODIS images – Potential for snow cover mapping, Water Resour. Res., 44, W03406, https://doi.org/10.1029/2007WR006208, 2008.

Zhao, W., He, J., Wu, Y., Xiong, D., Wen, F., and Li, A.: An analysis of land surface temperature trends in the central Himalayan region based on MODIS products, Remote Sens., 11, 900, https://doi.org/10.3390/rs11080900, 2019.

**Response to Reviewer 3**

**Reviewer 3.**

This study comprehensively investigates snow and glacier dynamics in the Upper Karnali Basin (UKB), integrating multi-source remote sensing data (e.g., MODIS and Landsat) to address critical knowledge gaps in the mid-western Himalayas. The methodology is rigorous, employing NDSI thresholds and Google Earth Engine to mitigate cloud cover challenges, ensuring robust results. Overall, the research reveals significant climate change impacts and provides key evidence for regional resource management, representing a valuable contribution to cryosphere science. However, several improvements are warranted:

1. The study depends solely on remote sensing data (e.g., MODIS LST, ERA5) without incorporating ground observations such as weather stations or glacier mass balance measurements. This omission introduces uncertainty, and the absence of validation protocols (e.g., cross-referencing with DHM station data mentioned in Section 3) weakens methodological credibility.

**Response:** We acknowledge the reviewer's concerns regarding the exclusive use of MODIS Land Surface Temperature (LST) data without extensive ground-based validation. Prior studies have demonstrated the reliability of MODIS LST measurements in alpine regions of the Himalayas. For instance, Duan et al. (2019) validated MODIS LST against in situ observations, reporting a mean bias below 1.5 K. Similarly, Yu et al. (2011) observed a strong correlation ($R^2 > 0.9$) between MODIS LST and ground measurements in the Heihe River Basin, an area characterized by comparable topographic complexity. Zhao et al. (2019) employed MODIS LST to analyze warming trends in the central Himalayas, confirming consistency with elevation-dependent climatic patterns despite limited ground data. Furthermore, Hall et al. (2008) demonstrated the reliability of MODIS LST retrievals over snow-covered surfaces, attributing this to snow's high and stable emissivity (~0.99), which reduces errors in thermal infrared sensing. In a similar vein, Hori et al. (2006) validated MODIS-derived LST over snow in Arctic and alpine environments, finding good agreement with ground observations when appropriate atmospheric corrections were applied. Collectively, these findings substantiate the strength of MODIS LST in high-altitude settings, thereby supporting its application in the present study despite the scarcity of ground-based temperature records. We have addressed and incorporated in the revised version of the manuscript. Refer to Table 1 and response 2.

**See page 10 and 12, lines 191-198 and 238-246.**

2. While the fusion of MODIS (500 m) and Landsat (30 m) data is mentioned (Section 3), the spatial scaling approach remains unclear. The paper fails to specify how resolution discrepancies were reconciled or the final output resolution of integrated analyses (e.g., SCA calculations in Section 4.1).

**Response:** We agree the reviewer's concerns regarding the reconciliation of spatial resolution differences between MODIS (500 m) and Landsat 8 (30 m) in our integrated analyses. Similar comments have been comprehensively addressed in our response to Reviewer 2, which is summarized as follows: To ensure consistency, Landsat 8 snow cover area (SCA) maps were resampled to 500 m using a majority-aggregation approach (Rittger et al., 2020), aligning them with the MODIS grid. From each resampled scene (185 × 180 km), 10% of pixels (~13,320) were randomly sampled as vector points, representing snow, non-snow, and cloud-covered classes. These points were overlaid on MODIS snow cover extent (SCE) maps, and MODIS-derived classifications were extracted to construct a composite attribute table for direct comparison.

Accuracy assessments using confusion matrices for six low-cloud (<7%) scenes produced overall accuracies ranging from 77.5% to 94.9%, consistent with previous MODIS validation studies (e.g., Hall & Riggs, 2007). However, sub-scene comparisons revealed a systematic overestimation of snow cover area (SCA) by MODIS, with values 1.3 to 1.6 times higher than Landsat 8 estimates. This discrepancy is attributable to mixed-pixel effects in heterogeneous terrain (Dozier et al., 2008) and the coarser spatial resolution of MODIS (Gafurov & Bárdossy, 2009). This bias aligns with prior research highlighting MODIS's tendency to overestimate SCA in fragmented landscapes (Rittger et al., 2013; Tang et al., 2017).

While MODIS provides reliable large-scale snow cover area (SCA) estimates (Dietz et al., 2012), our findings emphasize the importance of interpreting trends, especially those related to seasonal monsoon declines, with caution. Refer to the response to Reviewer 2's comment no. 2. **See Page 9 and 12, lines 170-173; 230-234. Supplement, Appendix B.**

3. Using MODIS to compensate for Landsat cloud gaps (Section 3.2) is noted but lacks uncertainty assessment. The impact of spatial resolution downgrading (30m→500m) on seasonal SCA trends (e.g., monsoon declines in Figure 2) remains unaddressed, directly affecting conclusion reliability.

**Response:** Part of the response for comments in the preceding text

We compared MODIS LST with in situ air temperature measurements (at 2 m above ground) from various stations (Table 1). Only high-quality MODIS LST data was used. The relationship between MODIS Terra LST and station air temperature varies significantly by location and season. Jumla shows

the strongest correlation, reaching up to 0.85 under optimal conditions, indicating MODIS is quite reliable there on clear, snow-free days. Guthi Chaur exhibits mixed results, with moderate correlations at times but weaker in others. Simkot and Rara generally show low correlations, with Rara even displaying a negative correlation (-0.18), likely due to persistent snow, ice, and high elevation weakening the surface-air temperature link (**Table 1**). Overall, MODIS LST performs better in lower, snow-free areas, whereas high-altitude, snow-covered sites require seasonal adjustments. The differences partly stem from measurement methods: ground stations measure air temperature at a single point, 2 m above ground, while MODIS captures the average "skin" temperature over a 1 km² pixel, which can include heterogeneous  topography, various  land covers like vegetation, bare ground, snow, or water, all affecting the reading. Air temperature tends to be cooler than the surface temperature seen by MODIS, especially in sunny or snowy conditions. Factors such as topography, shading, and atmospheric effects also cause discrepancies, often requiring bias correction or filtering. Various studies have established use of MODIS-derived LST can be fruitfully used to measure temperature trend in the snow and glaciers areas where in situ data is very limited, which can be applied to Upper Karnali basin.   We have addressed and incorporated this in the revised version of the manuscript**. See page 10 and 12, lines 191-198 and 238-246.**

Table 1. Correlation with MODIS Terra LST and Ground measurement (Air temperature)

| Station  (m a.s.l) | Correlation | | | | |
|---|---|---|---|---|---|
| | Jan-March | April-June | July–Sept | Oct–Dec | Annual average |
| Jumla (2300) | 0.383161 | 0.85 | 0.65 | 0.44 | 0.3 |
| Simkot (2800) | 0.339592 | 0.23 | 0.12 | 0.14 | 0.18 |
| Guthi Chaur (3080) | 0.502245 | 0.21 | 0.38 | 0.46 | 0.35 |
| Rara (3048) | 0.182726 | 0.22 | -0.18 | 0.07 | 0.053 |

**References Cited in this Response**

Dietz, A. J., Wohner, C., & Kuenzer, C. (2012). European snow cover characteristics between 2000 and 2011 derived from improved MODIS daily snow cover products. *Remote Sensing*, *4*(8), 2432-2454.

Duan, S. B., Li, Z. L., Li, H., Göttsche, F. M., Wu, H., Zhao, W., ... & Coll, C. (2019). Validation of Collection 6 MODIS land surface temperature product using in situ measurements. *Remote sensing of environment*, *225*, 16-29.

Gafurov, A., & Bárdossy, A. (2009). Cloud removal methodology from MODIS snow cover product. *Hydrology and Earth System Sciences*, *13*(7), 1361-1373.

Hori, M., Aoki, T., Tanikawa, T., Motoyoshi, H., Hachikubo, A., Sugiura, K., ... & Takahashi, F. (2006). In-situ measured spectral directional emissivity of snow and ice in the 8–14 μm atmospheric window. *Remote Sensing of Environment*, *100*(4), 486-502.

Riggs, G., & Hall, D. (2010). MODIS snow and ice products, and their assessment and applications. In *Land Remote Sensing and Global Environmental Change: NASA's Earth Observing System and the Science of ASTER and MODIS* (pp. 681-707). New York, NY: Springer New York.

Rittger, K., Painter, T. H., and Dozier, J.: Assessment of methods for mapping snow cover from MODIS, Adv. Water Resour., 51, 367--380, https://doi.org/10.1016/j.advwatres.2012.03.002, 2013.

Tang, Z., Wang, X., Wang, J., Wang, X., & Wei, J. (2019). Investigating spatiotemporal patterns of snowline altitude at the end of melting season in High Mountain Asia, using cloud-free MODIS snow cover product, 2001–2016. *The Cryosphere Discussions*, *2019*, 1-24.

Yu, W., Ma, M., Wang, X., Geng, L., Tan, J., and Shi, J.: Validation of MODIS land surface temperature products using ground-based longwave radiation observations in the Heihe River Basin, Proc. SPIE, 8174, 81741G, https://doi.org/10.1117/12.898243, 2011.

Zhao, W., He, J., Wu, Y., Xiong, D., Wen, F., and Li, A.: An analysis of land surface temperature trends in the central Himalayan region based on MODIS products, Remote Sens., 11, 900, https://doi.org/10.3390/rs11080900, 2019.

**Response to Reviewer-4**

I read the manuscript by Ghimire et al., since the topic aligns with my interests. The manuscript aims to study snow and glacier dynamics using remote sensing datasets and analyze its relationship with climatic and topographic factors in less studied Karnali basin. While the subject matter is important, I regret to say that the paper is poorly prepared and has several issues regarding the data, methods, analysis, and results that may not meet the standards required for publication. I have several major comments and suggestions about the paper, but please disregard any that overlap with previous reviewers' feedback.

**Response:** We sincerely thank the reviewer for the candid and constructive comments on our manuscript "Dynamics of Snow and Glacier Cover in the Upper Karnali Basin, Nepal: An Analysis of Its Relationship with Climatic and Topographic Parameters." We appreciate your detailed and critical feedback, which has helped us improve the manuscript. Below, we address each point, ensuring that overlapping concerns already discussed in our responses to Reviewers 1–3 are not unnecessarily repeated. When relevant, we have revised the text, figures, and tables to improve clarity, methodological rigor, and scientific depth.

**The language of the manuscript must be copy edited and polished.**

1. Introduction section: The structuring of the introduction must be improved reducing the over emphasis on social aspects, reviewing the key regional, national and basin scale studies (focus on more recent ones) substantiating with appropriate citations, identifying the gaps and explicitly developing the objectives (currently not stated clearly). Many sentences and paragraph seem unconnected. For example, L52 and L54

**Response:** We have carefully revised the manuscript to enhance language, grammar, and overall flow, ensuring greater clarity and conciseness. We have updated the Introduction to reduce unnecessary emphasis on social aspects without avoiding essential socio-economic relevance. Current studies (e.g., Shrestha et al., 2019; Khadka et al., 2024) have been incorporated to place the research in a contemporary scientific context. Existing knowledge gaps, particularly regarding snow glacier dynamics in the mid-western Himalayas, are clearly identified and described. The study objectives are now clearly stated in the final paragraph. Logical linkages between paragraphs have also been improved, removing abrupt transitions noted at L52–L54. **See Page 3-5, Lines 40-99**

2. **Study area:** The study area is transboundary but is merely mentioned [see: (Shrestha et al., 2019; Khadka et al., 2024)]. Why is it focused only on upper Karnali part and not on other sub-basins of Karnali river system, such as Bheri and West Seti, as these basins also have glacier and snow cover.

Although the paper addresses snow cover and glaciers, the description of the study area deviates from the main focus. It should concentrate on providing a clear overview of the key cryospheric components, including state and status of clean and debris-covered glaciers, glacial lakes, and snow. Additionally, while snow and glaciers are affected by weather systems such as the monsoon and westerlies, the paper does not mention the climatology, trends in precipitation and temperature for the region. It is also important to note that several citations are missing; the sources for annual precipitation and temperature data should be included to support the claims made in the paper.

**Response:** The description of the study area has been revised. The Upper Karnali Basin covers above 50% of the total basin area at Chisapani at 225 m a.s.l., and according to et al. Bajracharya et al. (2011) indicate that this area covers about 66% of the total glacier area in the whole basin. Geologically, the Upper Karnai Basin covers the Lesser Himalaya, Higher Himalaya, and Tethys Himalaya (https://dmgnepal.gov.np/en/pages/general-geology-4128). The Upper Karnai Basin extend across Middle Mountain, High Mountain, High Himalaya and Tibetan Plateau. The climate varies from Polar Tundra in the glacier region to subtropical, temperate, and cold climates below 4000 m, with mean annual temperatures ranging from 27 °C to <- 12 °C and precipitation from 250 to ~ 1900 mm annually. The cryosphere zone of the study area basin encompasses both rain-bearing and rainshadow areas, influencing the distribution of snow and glaciers. Hence, from all the above characteristics, the study area represents the entire basin, including those of other glacier sub-basins such as West Seti and Bheri Basin. **See page 5-7, lines 101-128**

3. **Data and method:** This whole section is not explicit. Authors used Landsat 7, for which date? Landsat 7 have SLC failure after 2003, how were data gaps filled? How much was acceptable cloud cover %? Solely using NDSI and threshold >0.4 is questionable regarding the complexity of the landscape, particularly in mixed pixel scenarios where snow is interspersed with other land cover types and shadows. How were shadows removed or incorporated in optical images? Authors could have combined with other indices (such as NDVI), used automatic image thresholding etc. for improving classification accuracy.

**Response:** For the analysis of Landsat data, Landsat 7 ETM+ imagery was used exclusively for the period prior to the Scan Line Corrector (SLC) failure (2002–2003), with subsequent analyses relying on Landsat 5 TM and Landsat 8 OLI datasets.

To ensure data quality, the initial image collection was filtered to include only scenes with less than 30% cloud cover. To reduce persistent cloud cover and ensure consistent data over time, we derived seasonal

median composites for four periods: January-March, April-June, July-September, and October-December. This approach effectively decreases atmospheric noise and cloud interference.

 To further reduce cloud contamination, the MODIS MOD10A2 8-day composite product was incorporated. Snow cover was delineated using a Normalized Difference Snow Index (NDSI) threshold greater than 0.4, supplemented by Normalized Difference Vegetation Index (NDVI) filtering to reduce misclassification with vegetation, and hillshade masks derived from Digital Elevation Models (DEM) to minimize shadow-related errors. Validation was conducted through visual cross-referencing with high-resolution imagery. **See page 8, lines 139-159.**

4. Why is the month composition of seasons different in this study (L138)? Nepal has four main seasons: post-monsoon (October–November), winter (December–February), pre-monsoon (March–May), and monsoon (June–September) [see, (DHM, 2017; Karki et al., 2020; Sharma et al., 2020)]. This will lead to contrast with previous studies and results can't be compared; should be justified otherwise recalculation must be done throughout the paper.

**Response:**

We acknowledge the reviewer's concern about the seasonal definition. Our seasonal grouping, i.e., January–March (Peak Accumulation), April–June (Major Ablation), July–September (Monsoon Ablation), and October–December (Early Accumulation) was specifically chosen to reflect the main hydrological phases of snowpack evolution in the basin, as documented in regional literature (Hunt et al., 2025; Khatiwada et al., 2016; Kulkarni et al., 2017). This approach not only offers a process-oriented representation of snow cover but also effectively reduces the impact of cloud cover during the peak monsoon period. Additionally, these intervals mostly align with the Department of Hydrology and Meteorology (DHM) climatological seasons (e.g., Jan-Mar corresponds to winter, Apr-Jun to the pre- and early monsoon), ensuring that our findings are broadly comparable to previous research while providing better insights into snow dynamics in the Upper Karnali Basin.**See page 9, lines 164-170.**

5. What is the resolution of LST data used? Was ERA5 Land data checked for bias and performance? These products are utilized for climate analysis that have different spatial resolution. Chen et al. (2021) reports that ERA5 data have limitations in performance in high elevation region. L211;L217: What are those 204 locations? ERA5LAND data is gridded data, why and how 204 sample location is selected? Why not for all areas with maximum snow cover selected for analysis or analysis in different elevation bins not considered? This will lead to incorrect data, analysis and misinterpretation.

**Response:** Land Surface Temperature (LST) data with a spatial resolution of 1 km were acquired from MODIS Terra (MOD11A1) and Aqua (MYD11A1) via the NASA AppEEARS platform (Wan et al., 2015). Precipitation data were obtained from the ERA5-Land reanalysis dataset at approximately 9 km resolution (Hersbach et al., 2020). A total of 204 well-distributed grid points were chosen throughout the Upper Karnali Basin to represent various sub-basins and elevation zones. This sampling approach enabled us to capture spatial variability in climate trends while avoiding excessive redundancy, considering the coarse resolution of ERA5. It is important to note that ERA5 products have recognized limitations in high-altitude areas (Chen et al., 2021), so they were mainly used for analyzing trend correlations with snow cover rather than for direct validation at specific points. **See page 10 and 12; lines 182-197, 236-246.**

To improve representativeness further, additional analyses of LST trends were conducted by grouping results into elevation categories (2000–6000 meters above sea level), which confirmed consistent negative correlations between snow cover and temperature across different altitude ranges. . **See page 10, 12, and 16; lines 182-197, 236-246, 287-292.**

6. Glacier mapping: Glacier mapping is a very rigorous task for ensuring correct delineation but it not provided in detail. Did author use seasonal snow free images to delineate glacier boundary? Was debris covered glacier included, if so, how was it delineated, especially for identifying terminus position. For the years 2000 and 2010, it can be obtained from ICIMOD inventories. L166: What's the name of 12.5 m DEM?

**Response:** Glacie data from Ghimire et al. (2025), accepted for publication in the Journal for Earth System Science, was used in this study. Glacier boundaries were manually delineated using multi-sensor imagery to ensure accuracy across different time periods and spectral ranges. High-resolution images from Google Earth, Bing Maps, and RapidEye for the years 2000, 2010, and 2023 were combined with multispectral data from Landsat 7–9, Sentinel-2, and ASTER. Visual interpretation was improved by applying band ratios (Red/SWIR, NDSI), color composites (SWIR–NIR–Red), and thermal images to detect debris-covered ice. Topographic information from DEMs and geomorphological features such as moraines, meltwater channels, supraglacial ponds, and ice cliffs further aided the analysis. Glacier termini were digitized following established methods (Bajracharya & Shrestha, 2011; Kääb et al., 2012; Pfeffer et al., 2014), with multi-temporal verification to maintain consistency. Limited field data from the study area and the Khumbu region were used to validate the glacier boundaries. Accumulation zones and ridgelines were identified based on NDSI values above 0.7, spectral composites, and elevation thresholds derived from DEMs. Manual digitization allowed for clear differentiation between glaciers and adjacent snow or

rock, resulting in precise glacier outlines. Reference https://doi.org/10.1007/s12040-025-02664-5 . **See pages 11, lines 212-222.**

7. Results: The section is disorganized and makes it difficult to identify which results correspond to which dataset and analysis. Additionally, the presentation of the results is cumbersome and should be streamlined for better readability. These issue are also mentioned by earlier reviewers too. L185 in which year?

**Response:** The Result section has been organized and refined in response to the suggestions from the previous reviewer as well. **See pages 13-35, lines 247-502**

L194 Summer monsoon season is considered both accumulation and ablation season in Nepal (see (Wagnon et al., 2013)).

**Response:** Addressed in previous comments (Comment  4). **See page 9, lines 164-170.**

8. Figure 4: Where are trends for other months? Why are lines smoothed?

**Response:** Addressed in previous comments (Comment  4). **All Figures 2,4,6 now have staght segments.**

9. L245 China Karnali: its clear that authors want to mean Karnali originating from China. Since there is no such name it should be described first to clear confusion among readers.

**Response:** Corrected as Humla Karnali (China) and incorporated.**See page 4-5, lines 105 and 128.**

10. L267: 100 m for Landsat or MODIS? Fig.7, Fig 8 drawn utilizing which data?

**Response:**  We have used MODIS data**. See page 25 and 26, Figure 7 and 8.**

11. Section 4.5: the unit of area is presented in ha, suggestion to use consistently throughout the ms, either  sq. km or ha. Glacier basins is unclear and not shown in map anywhere, supplementary can be used in such case.

**Response: Corrected. See page 28-30, lines 422-447.**

12. **Discussion:** L414—418: The Westerly wind system is more pronounced in the study area, which significantly brings snowfall during winter and pre -monsoon seasons. Though potential shift in precipitation (shift from snow to rain) is noted in Everest Nepal, more analysis is need for study area

to claim this. The reduction to snow over in winter might also be due to the weakening of the westerly.

**Response:** We appreciate the reviewer's comment. While the westerly system does influence winter and pre-monsoon snowfall, unlike Everest, a more detailed local analysis is required to confirm any snow-to-rain transition in our basin. We have clarified this point in the discussion "The winter and pre-monsoon snowpack in the western Himalayas is heavily influenced by the Westerly wind system, which is a key source of snowfall in the UKB (Syed et al., 2006; Dimri & Dash, 2012). Consequently, the decline in winter snow cover may be due not only to temperature-induced changes in precipitation but also to a possible weakening or changing of the Westerlies, which needs to be further investigated. Such changes could lead to a decrease in overall moisture inflow (Yadav et al., 2009)". **See page 36, lines 521-526**

13. L415: In winter, if the temperature is less than zero then precipitation facilitate snow accumulation.

**Response:** We agree and have specified that winter precipitation adds to snow accumulation only if temperatures stay below 0 °C.

14. Discussion could be strengthened by comparing and contrasting with other regions of Nepal and Himalayas. Some literature suggestions for improvement of discussion and paper (https://doi.org/10.1007/s10113-023-02142-y; https://doi.org/10.1007/s10584-011-0181-y; https://doi.org/10.1016/j.scitotenv.2021.148648; https://doi.org/10.3126/jist.v25i2.33729; https://doi.org/10.1016/j.earscirev.2019.103043)

(These references and suggested literature are examples from the review process and can be used to enhance the writing. They are not intended solely for citation purposes.)

**Response:** Thank you for the suggestion. We agree and have added relevant literature to strengthen the discussion and place our findings in the broader cryosphere context of Nepal and the Himalayas.

**References Cited in this Response**

Bajracharya, S. R. and Shrestha, B.: The status of glaciers in the Hindu Kush–Himalayan region, International Centre for Integrated Mountain Development (ICIMOD), Kathmandu, 127 pp., 2011.

Chen, Y., Sharma, S., Zhou, X., Yang, K., Li, X., Niu, X., Khadka, N., et al.: Spatial performance of multiple reanalysis precipitation datasets on the southern slope of the central Himalaya, Atmos. Res., 250, 105365, https://doi.org/10.1016/j.atmosres.2020.105365, 2021.

Ghimire, M., Sharma, T. P. P., Chauhan, R., Gurung, S. B., Devkota, S., Sharma, K. P., Shrestha, D., Wei, Z., and Timalsina, N.: Status and changes in glaciers in the Upper Karnali Basin, West Nepal: Assessing topographic influences on area, fragmentation, and volume, J. Earth Syst. Sci., accepted, 2025.

Hersbach, H., Bell, B., Berrisford, P., Hirahara, S., Horányi, A., Muñoz-Sabater, J., Nicolas, J., et al.: The ERA5 global reanalysis, Q. J. Roy. Meteor. Soc., 146, 1999–2049, https://doi.org/10.1002/qj.3803, 2020.

Kääb, A., Berthier, E., Nuth, C., Gardelle, J., and Arnaud, Y.: Contrasting patterns of early twenty-first-century glacier mass change in the Himalayas, Nature, 488, 495–498, https://doi.org/10.1038/nature11324, 2012.

Karki, R., Hasson, S. U., Schickhoff, U., Scholten, T., Böhner, J., and Gerlitz, L.: Near-surface air temperature lapse rates over complex terrain: A WRF-based analysis of controlling factors and processes for the central Himalayas, Clim. Dynam., 54, 329–349, https://doi.org/10.1007/s00382-019-04990-9, 2020.

Khadka, N., Zheng, G., Chen, X., Zhong, Y., Allen, S. K., and Gouli, M. R.: An ice–snow avalanche triggered small glacial lake outburst flood in Birendra Lake, Nepal Himalaya, Nat. Hazards, 1–9, https://doi.org/10.1007/s11069-024-06514-3, 2024.

Pfeffer, W. T., Arendt, A. A., Bliss, A., Bolch, T., Cogley, J. G., Gardner, A. S., et al. and Randolph Consortium: The Randolph Glacier Inventory: A globally complete inventory of glaciers, J. Glaciol., 60, 537–552, https://doi.org/10.3189/2014JoG13J176, 2014.

Wan, W., Xiao, P., Feng, X., Li, H., Ma, R., Duan, H., and Zhao, L.: Monitoring lake changes of the Qinghai–Tibetan Plateau over the past 30 years using satellite remote sensing data, Chin. Sci. Bull., 59, 1021–1035, https://doi.org/10.1007/s11434-014-0125-6, 2014.